# UVR2 ensures transgenerational genome stability under simulated natural UV-B in *Arabidopsis thaliana*

Eva-Maria Willing[1,*], Thomas Piofczyk[2,*], Andreas Albert[3], J. Barbro Winkler[3], Korbinian Schneeberger[1] & Ales Pecinka[2]

Ground levels of solar UV-B radiation induce DNA damage. Sessile phototrophic organisms such as vascular plants are recurrently exposed to sunlight and require UV-B photoreception, flavonols shielding, direct reversal of pyrimidine dimers and nucleotide excision repair for resistance against UV-B radiation. However, the frequency of UV-B-induced mutations is unknown in plants. Here we quantify the amount and types of mutations in the offspring of *Arabidopsis thaliana* wild-type and UV-B-hypersensitive mutants exposed to simulated natural UV-B over their entire life cycle. We show that reversal of pyrimidine dimers by UVR2 photolyase is the major mechanism required for sustaining plant genome stability across generations under UV-B. In addition to widespread somatic expression, germline-specific UVR2 activity occurs during late flower development, and is important for ensuring low mutation rates in male and female cell lineages. This allows plants to maintain genome integrity in the germline despite exposure to UV-B.

[1] Department of Plant Developmental Biology, Max Planck Institute for Plant Breeding Research, Carl-von-Linné-Weg 10, D-50829 Cologne, Germany. [2] Department of Plant Breeding and Genetics, Max Planck Institute for Plant Breeding Research, Carl-von-Linné-Weg 10, D-50829 Cologne, Germany. [3] Research Unit Environmental Simulation, Helmholtz Zentrum München, Ingolstädter Landstrasse 1, D-85764 Neuherberg, Germany. * These authors contributed equally to this work. Correspondence and requests for materials should be addressed to K.S. (email: schneeberger@mpipz.mpg.de) or to A.P. (email: pecinka@mpipz.mpg.de).

Plants require sunlight for photosynthesis and developmental regulation[1]. However, ground levels of solar radiation also contain a low proportion of UV-B radiation (UV-B, 280–315 nm), which has multiple effects on plants including photomorphogenic and damaging responses[2–4]. Photomorphogenic responses are triggered upon UV-B perception by UV-B-RESISTANCE 8 (UVR8)[2,5]. UV-B-irradiated UVR8 homodimers will monomerize and bind COP1 E3 ubiquitin ligase. Reduced COP1 activity will allow accumulation of HY5 transcription factor and will trigger UV-B transcriptional response of ~100 target genes and more compact plant growth, including, e.g., reduced plant height and shorter petioles[4]. Furthermore, low UV-B levels boost accumulation of flavonoid pigments, in a TRANSPARENT TESTA 4 (TT4)-dependent manner, which will build up a protective sunscreen layer contributing to UV-B acclimation and even protection against other stresses[5,6]. Higher natural and, in particular, laboratory-applied UV-B doses cause damage[3,7,8]. This involves a burst of reactive oxygen species, damages to cell membranes, proteins and DNA. The major types of UV-B-induced DNA damage are pyrimidine dimers and, to a lower extent, also DNA strand breaks[9–11]. Pyrimidine dimers are non-native bonds between two pyrimidines (cytosine and thymine). They disturb DNA structure, interfere with replication and transcription, and are therefore generally repaired[12]. The cyclobutane pyrimidine dimers (CPDs; 75–90% of all pyrimidine dimers) and 6,4 pyrimidine-pyrimidones ((6-4)PPs; 10–25% of all pyrimidine dimers) are directly reverted by UV-B-RESISTANCE 2 (UVR2) and UV-B-RESISTANCE 3 (UVR3) photolyases, respectively, in somatic tissues[13,14]. An alternative repair pathway common to all eukaryotes involves nucleotide excision repair (NER). In A. thaliana, loss of NER-associated endonuclease UV-B HYPERSENSITIVE 1 (UVH1), an orthologue of human XERODERMA PIGMENTOSUM COMPLEMENTATION GROUP F (XPF), leads to failures in repair of UV-B-induced lesions and reduced growth in response to UV-B treatment[15,16]. Owing to the low UV-B penetration into plant tissues through flavonoid layer[17], most of the UV-B-induced mutations are to be expected in the epidermal cells. However, there is some evidence that UV-B may penetrate also into deeper meristematic cell layers as even low UV-B increases genome instability in the plant germline[11]; however, the precise frequencies of UV-B-induced mutations and their molecular spectra remain unknown in plants.

Here we determined mutation frequencies in germline DNA of A. thaliana wild-type and UV-B-hypersensitive mutants exposed to UV-B treatment by a combination of whole-genome sequencing and genetic analyses. We found that mutations induced by the UV-B treatment have specific spectra, preferentially occur in particular sequence contexts and have other characteristics that differentiate them from spontaneous mutations. Furthermore, we show that direct reversal by UVR2 photolyase is the key pathway limiting the frequency of UV-B treatment-induced mutations in the DNA of germline cells. We localized this repair activity into late flower development after the split of male- and female-specific cell lineages.

## Results

### Effects of simulated solar UV-B on A. thaliana growth.
Wild-type plants and six mutant genotypes uvr8, tt4, uvh1, uvr2, uvr3 and uvr2 uvr3 found as UV-B- and/or UV-C-hypersensitive in previous studies[6,15,18,19] were cultivated during their entire life cycle in sun simulators[20] for up to three generations without UV-B (hereafter as 'control') and with a biologically effective UV-B radiation (UV-B$_{BE}$) normalized at 300 nm (ref. 21) of 100, 150 and 300 mW m$^{-2}$ (Fig. 1a and Supplementary Fig. 1a–c). Owing

to the filtering conditions used, this UV-B treatment did lead to more UV-A than in the control treatment. However, the amount of UV-A radiation in the control treatment reached up to 80% and more for wavelengths greater than 360 nm compared to the UV-B treatments. Below 360 nm the transmission decreased due to the transmission characteristics of the filter glass, therefore, the UV-A radiation is reduced to about 10% at 330 nm compared to the UV-B treatments. The UV-B treatments resembled natural conditions during the main A. thaliana-growing season (April/May) along the European north-south UV-B cline at 60°N, 52°N and 40°N, which can be approximated to Helsinki, Berlin, and Madrid, respectively. Wild-type and all mutant genotypes showed comparable growth at rosette stages under control conditions (Fig. 1b). Under the highest simulated natural UV-B, wild-type and uvr8 plants did not show significantly reduced rosette diameter, while tt4, uvr2, uvr3, uvr2 uvr3 and uvh1 mutant plants did (t-test P values: 5.390E−01, 9.113E−01, 4.3E−06, 1.6E−16, 4.4E−02, 2.6E−16 and 8.2E−03, respectively; Fig. 1b). This suggested that not all A. thaliana mutants found to be UV-B- and/or UV-C-hypersensitive in laboratory would show similar phenotypes under natural UV-B conditions.

### Frequency of mutations induced by UV-B treatment.
The seeds of control and UV-B-treated plants were grown under non-UV-B conditions and whole genomes of 146 offspring plants, typically five per genotype and treatment, were sequenced (Supplementary Fig. 2 and Supplementary Data 1). This revealed a total of 2,497 novel single-base substitutions and 22 one-to-four base pair deletions. Using di-deoxy sequencing, we confirmed 58 out of 59 randomly selected mutations, suggesting a 1.7% false-positive discovery rate in our analysis (Supplementary Data 2 and Methods). A false-negative mutation discovery rate was estimated to be 0.15% by simulations (see Methods).

Wild-type plants without UV-B treatment accumulated on average 2.6, 2.0 and 2.4 spontaneous mutations per haploid genome and generation (hereafter as 'mutations') in the first (Fig. 1c), the second and the third generations (generation average 2.3), corresponding to 2.2, 1.7 and 2.0 × 10$^{-8}$ mutations per site, respectively (Supplementary Data 1). Similar numbers of novel mutations (2.0–5.7) were observed in the progenies of control uvr8, tt4, uvr2, uvr3 and uvr2 uvr3 plants (Fig. 1c and Supplementary Data 1). In contrast, compromised NER in uvh1 plants resulted in 20.3 mutations. This represented 7.8-fold increase (Fisher's exact test, P = 4.9E − 12) compared with wild-type and illustrated importance of NER for general genome stability in A. thaliana.

Treatment with 100, 150 and 300 mW m$^{-2}$ induced 3.3, 5.0 and 2.8 mutations, respectively, per haploid genome and generation in wild-type plants (Supplementary Fig. 3a). Subsequently, the UV-B$_{BE}$ of 300 mW m$^{-2}$ was used as the standard UV-B treatment. Loss of UVR8 and TT4 functions did not significantly change the mutation rates (5.6 versus 7.8 and 5.7 versus 6.7 mutations under control and UV-B; Fisher' exact test P = 0.2203 and 0.6455, respectively; Fig. 1c). In UV-B-treated uvh1 plants, we found 27.4 new mutations, which represented a significant 1.3-fold increase compared with 20.3 new mutations under control conditions (Fisher's exact test, P = 0.03772).

The only drastic increase in mutation rate in a single mutant was observed in the progeny of UV-B-irradiated uvr2 plants containing on average 64.3 new mutations (Fig. 1c). This corresponded to a high 14.7-fold increase over the control uvr2 plants with 4.4 mutations per genome and generation (Fisher's exact test, P < 2.2E − 16). The 7.3 new mutations in UV-B-treated uvr3 plants represented a lower, but still significant 2.1-fold increase over the control treatment (Fisher's exact test,

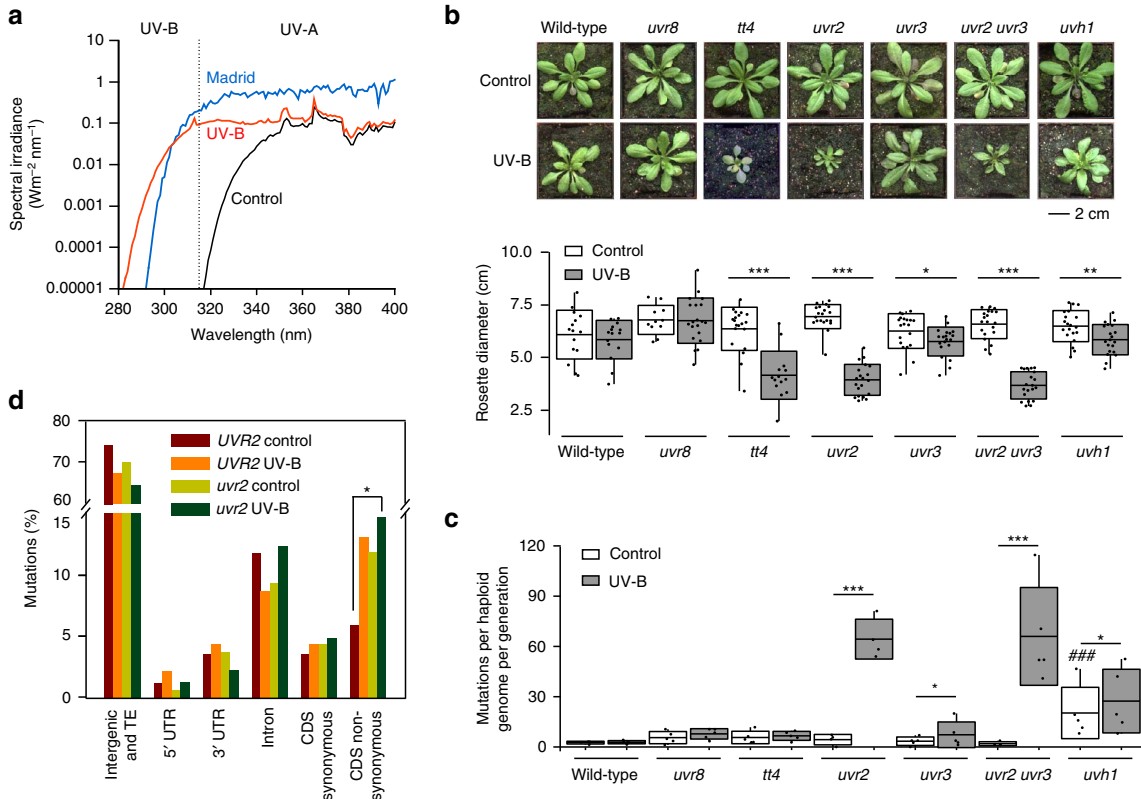

**Figure 1 | Frequencies and spectra of UV-B treatment-induced mutations.** (**a**) Spectral irradiance in sun simulator of the UV-B-free control (black; UV-B$_{BE}$ normalized at 300 nm (ref. 21) = 0 mW m$^{-2}$), and the simulated UV-B level of Madrid (red; UV-B$_{BE}$ normalized at 300 nm (ref. 21) = 300 mW m$^{-2}$) in UV-B and UV-A range (divided by dotted vertical line). The modelled Madrid UV-B$_{BE}$ (blue; UV-B$_{BE}$ normalized at 300 nm (ref. 21) = 265 mW m$^{-2}$) was generated using the Quick Tropospheric UV Radiation Calculator. (**b**) Representative phenotypes of individual genotypes grown under control and 300 mW m$^{-2}$ UV-B$_{BE}$. Rosette diameter measurements were performed on 11–20 plants per genotype and treatment. Significant differences in Student's *t*-test (*$P < 0.05$, **$P < 0.01$, ***$P < 0.001$). (**c**) Normalized number of control and mutations induced by UV-B treatment per haploid genome and generation. Boxes show genotype average (middle line), s.d. (lower and upper margins), and values outside of the s.d. range (vertical bars). Dots represent individual genomes. ### and * indicate statistically significant (*$P < 0.05$, **$P < 0.01$, ***$P < 0.001$) differences in Fisher's exact test between: ###mutants versus wild-type and *control versus UV-B treatments (300 mW m$^{-2}$ UV-B$_{BE}$) of the same genotype, respectively. (**d**) Frequency of non-synonymous amino-acid changes in different genomic regions. UVR2 includes Col-0, *uvr8*, *tt4* and *uvr3* genotypes treated with 0 mW m$^{-2}$ UV-B$_{BE}$ (control) or with 100, 150 and 300 mW m$^{-2}$ UV-B$_{BE}$ (UV-B). *uvr2* includes *uvr2* and *uvr2 uvr3* genotypes treated as control and UV-B. Numeric values are provided in Supplementary Table 1. *statistically significant (*$P = 0.0254$) difference in Fisher's exact test. All other comparisons within groups were not significant.

$P = 0.01965$). UV-B-exposed *uvr2 uvr3* double-mutant plants had 66.0 new mutations (Fisher's exact test, $P \leq 2.2E - 16$; Fig. 1c). The progeny of *uvr2 uvr3* plants exposed to 0, 100, 150 and 300 mW m$^{-2}$ UV-B$_{BE}$ revealed on average 2.0, 39.1, 65.3 and 66.0 mutations per haploid genome and generation, respectively (Supplementary Fig. 3b). This corresponded to 19.5-, 32.6- and 33-fold increase and indicated a UV-B dose-dependent accumulation of mutations at the lower and saturation at the higher UV-B doses, respectively (Fisher's exact test; all $P < 2.2E - 16$ in UV-B versus control; UV-B$_{BE}$ of 100 versus 150 and 300 mW m$^{-2}$: $P = 2.0E - 08$ and $1.2E - 08$; UV-B$_{BE}$ of 150 versus 300 mW m$^{-2}$: $P = 0.8978$).

The UV-B treatment also affected the frequency of non-synonymous amino-acid mutations. They were approximately threefold more frequent in UV-B-treated (300 mW m$^{-2}$ UV-B$_{BE}$) *uvr2* versus control wild-type plants (14.7% versus 5.9% of all mutations, respectively; Fisher's exact test $P = 0.0254$; Fig. 1d). In absolute terms, this corresponded to 10.2 new non-synonymous amino-acid mutations per one *uvr2* plant, compared with an average of 0.2, 0.4 and 0.5 such mutations in control wild-type, control *uvr2* and UV-B-treated wild-type plants, respectively (Supplementary Table 1). We also found phenotypically distinct plants in the third UV-B-irradiated generation of the

double mutant (see example of semidominant mutant in Supplementary Fig. 3c), suggesting an increased functional impact of the mutations induced by the UV-B treatment on gene integrity in *UVR2*-defective plants.

**Spontaneous and induced mutation spectra in *A. thaliana*.** To characterize the treatment-specific mutation spectra, we compared mutations from all control plants with those of all UV-B-treated plants with exception of *uvh1* samples, which were excluded owing to a 35% rate of A:T→T:A transversions, compared with <10% in the other genotypes (Supplementary Fig. 4a).

Consistent with previous observation of Ossowski *et al.*[22], about half (52%) of all substitutions under UV-B-free conditions were G:C→A:T nucleotide transitions (Fig. 2a). The G:C→A:T frequency increased to 88% after UV-B treatment (Fisher's exact test $P < 2.2E - 16$), which led to significantly reduced proportion of all other substitution types (Fig. 2a; Fisher's exact test $P$ values for control versus UV-B; A:T→G:C, 2.0E − 02; A:T→T:A, 9.6E − 05; G:C→T:A, 2.1E − 05; A:T→C:G, 3.9E − 12; G:C→C:G, 1.3E − 03). Therefore, simulated natural UV-B caused almost exclusively G:C→A:T nucleotide transitions.

To test whether this holds true in major genome fractions, we quantified mutation spectra in genes and transposons separately (Supplementary Fig. 4b). Under control conditions, G:C→A:T nucleotide transitions remained the major type of change in transposons (66%); however, this trend was absent in genes (23%)

where all six possible substitution types showed relatively similar frequencies (10–23%). We also observed more G:C→A:T nucleotide transitions in transposons (65%) than in genes (42%) within the data of Ossowski et al.[22] (Supplementary Fig. 4c). Surprisingly, after UV-B treatment, the G:C→A:T transition rate changed and was even larger in genes than in transposons (93% versus 87%; Fisher's exact test, P value = 0.0038; Supplementary Fig. 4b). Hence, transposons were prone to G:C→A:T transitions under both control and UV-B conditions, while genes only during UV-B treatment.

To find whether spontaneous mutation and those induced by UV-B treatment occurred in a particular sequence context, we performed a motif analysis around mutated sites. This revealed an absence of any specific mutation-prone context in the vicinity of spontaneously mutated G:C→A:T sites in control samples (Fig. 2b). However, within UV-B-treated plants C→T and G→A mutations occurred preferentially within the TC(C/T) and (G/A)GA contexts, respectively. Such an asymmetric and reverse complementing pattern strongly suggests that: (i) G→A mutations are C→T mutations on the reverse strand; (ii) mutations induced by the UV-B treatment occur predominantly at the 3′ base of the pyrimidine dimer; and (iii) that TC(C/T) represents the UV-B-mutation-prone sequence in A. thaliana.

**DNA methylation overlaps with the mutated sites.** On the basis of the preferential UV-B mutagenesis of DNA-methylated cytosines in the CpG context in mammals[23,24], we tested for correlation between DNA methylation patterns and mutations induced by the UV-B treatment in A. thaliana. Because DNA methylation is a very stable epigenetic modification, we used existing genome-wide DNA methylation data sets[25,26]. According to the functional types of DNA methylation in plants[25], we classified cytosines in the CG, CHG and CHH sequence contexts (where H is A, T or C) as being either methylated or non-methylated and scored for the methylation status at mutated

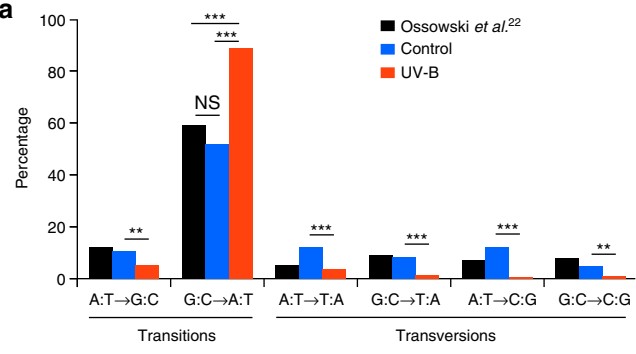

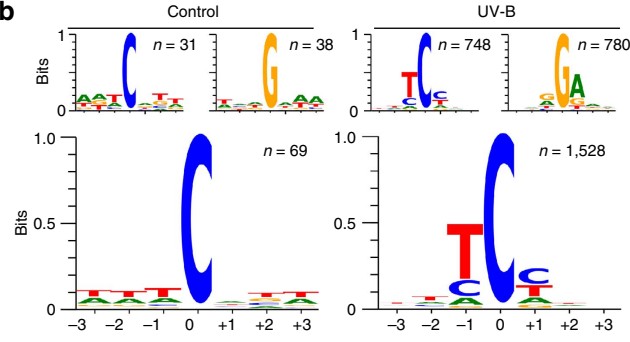

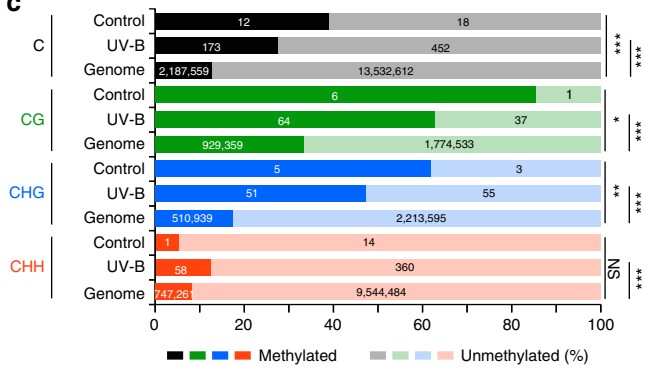

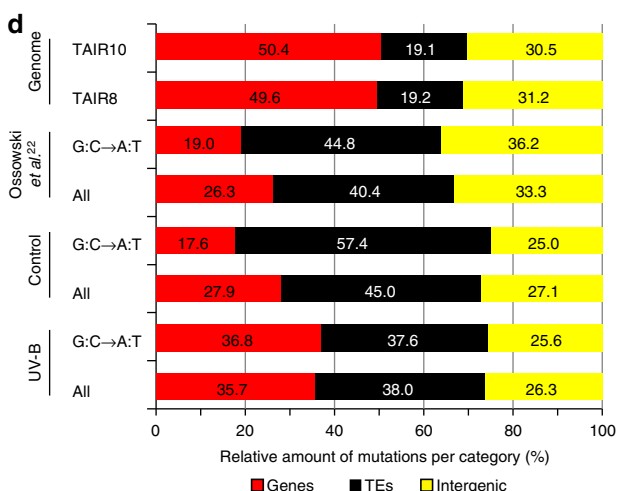

**Figure 2 | Genomic features of mutated positions.** (**a**) Proportions of single-nucleotide changes in Ossowski et al.[22] control samples (includes all genotypes treated with 0 mW m$^{-2}$ UV-B$_{BE}$; uvh1 was excluded) and UV-B-treated samples (includes all genotypes treated with 100, 150 and 300 mW m$^{-2}$ UV-B$_{BE}$; uvh1 was excluded). Statistical significance in Fisher's exact test: *P < 0.05, **P < 0.01, ***P < 0.001, n.s. = not significant. (**b**) DNA sequence motifs associated with control and mutations induced by UV-B treatment. Top images show cytosine and guanine mutation contexts on the forward strand. Bottom images show integrated information from both strands. Stacks' height indicates the sequence conservation measured in bits[44]. Symbol of mutated base at the position 0 was size reduced from 2 to 1 bit to reduce graph height. Height of other bases was not changed. Genomes are grouped into control and UV-B samples as described in **a**. (**c**) Percentage (x axis) of overlap of mutated positions with DNA methylation, and genome-wide DNA methylation frequencies for cytosines in C, CG, CHG and CHH contexts (where H is A, T or C). Values in columns show absolute number of mutated (Control and UV-B) or genomic positions (Genome) with available DNA methylation information. Statistical significance in Chi-square test with Yates correction: *P < 0.05, **P < 0.01, ***P < 0.001, n.s. = not significant. None of the control versus UV-B comparisons was significantly different (P > 0.05). Control samples were grouped as described in **a**. UV-B contained also 300 mW m$^{-2}$ UV-B$_{BE}$ samples. (**d**) Percentage of mutations in major genome fractions. A. thaliana genome composition according to TAIR8 and TAIR10 annotations. Proportions of spontaneous (Ossowski et al.)[22] control sun simulator and UV-B-treatment-induced (300 mW m$^{-2}$ UV-B$_{BE}$) mutations in genes, transposable elements (TE) and intergenic regions. Groups were analysed as 'all' mutations and G:C→A:T mutations only. Individual genotypes were grouped into control and UV-B samples as described in **a**.

positions. This revealed that both spontaneous and induced mutations overlapped with methyl-cytosines (with the exception of the CHH control group, which contained only 15 testable positions) significantly more often than expected at random based on the genome-wide DNA methylation frequencies (Chi-square test with Yates correction, $P$ values for control versus genome and UV-B versus genome: CNN: $1.12E-04$ and $<2.2E-16$; CG: $1.38E-02$ and $<2.2E-16$; CHG: $6.59E-03$ and $<2.2E-16$; CHH: $6.83E-01$ and $3.10E-07$; Fig. 2c). Hence, this suggests that methyl-cytosine is prone to mutate under UV-B conditions compared with non-methylated cytosine.

Because DNA methylation is concentrated into transposon-rich chromosomal regions in *A. thaliana*[25,26], we tested whether the mutations show particular genomic distribution. Both control and UV-B treatments led to hypo-accumulation of mutations in genes, relatively random accumulation in intergenic regions and hyper-accumulation in transposons (Fig. 2d). We confirmed this trend using independent data set of Ossowski *et al.*[22] However, UV-B treatment induced ~10% more mutations in genic regions compared with control plants. Therefore, the UV-B treatment adds to the mutagenic effect of DNA methylation, but also affects non-methylated cytosines in genic regions.

**Accumulation of induced mutations during development.** Early embryonic separation of gametic and somatic cell lineages largely prevents transgenerational inheritance of somatic mutations in mammals[27]. In contrast, the late separation of germline cells in plants[28] allows the inheritance of mutations induced during vegetative growth in cells of the apical meristem into the progeny. Alternatively, mutations can occur later after separation of male and female cell lineages and/or gamete formation. To determine whether mutation induced by UV-B treatment accumulated during particular developmental stages, we analysed the ratio of heterozygous and homozygous mutations in the progeny of the first generation of plants in control and UV-B treatments. If all mutations occurred before the differentiation of the male and female organs, we expected a 2:1 ratio of heterozygous versus homozygous mutations in an inbreeding constitutively monoecious species such as *A. thaliana*. We found ratios of 1.4:1 (wild-type control), 2.5:1 (wild-type UV-B-treated) and 1:1 (*uvr2* control), but there were significantly 8.1-fold more heterozygous than homozygous mutations (44.22 versus 5.44 per haploid genome, respectively) in the progeny of UV-B-treated *uvr2* plants (Fisher's exact test $P$ values when compared with the other groups: $2.95E-08$, $5.83E-05$ and $7.97E-05$, respectively; Fig. 3a). This strongly suggested that the combination of UV-B treatment with *uvr2* genotype leads to mutations mostly after the split of female and male cell lineages. To validate this, we expressed luciferase-tagged UVR2 under control of its native promoter (*UVR2promoter::UVR2:LUCIFERASE*). The reporter line showed strong UV-B-independent developmentally controlled UVR2 accumulation in meristems (root apical meristem, young leaves, flowers, flower buds, axillary buds, closed anthers and young pistils), scars after petals and sepals and weaker expression in expanded leaves (Fig. 3b–e; the control non-transgenic plants are shown in Supplementary Fig. 5). No expression was observed in green or dry seeds (Fig. 3e). The strong UVR2 expression in floral tissues supported the results of our genetic analysis.

Occurrence of a high number of mutations in male and female cell lineages allowed us to test whether there are sex-specific preferences in mutation accumulation in *A. thaliana*. We grew *uvr2 uvr3* plants under control UV-B-free conditions until bolting, and then exposed half of the plants to UV-B until flowering and subsequently reciprocally crossed UV-B-irradiated and control plants (Fig. 3f). The resulting F1 hybrids were grown under non-UV-B conditions, and genomes of eight plants per crossing

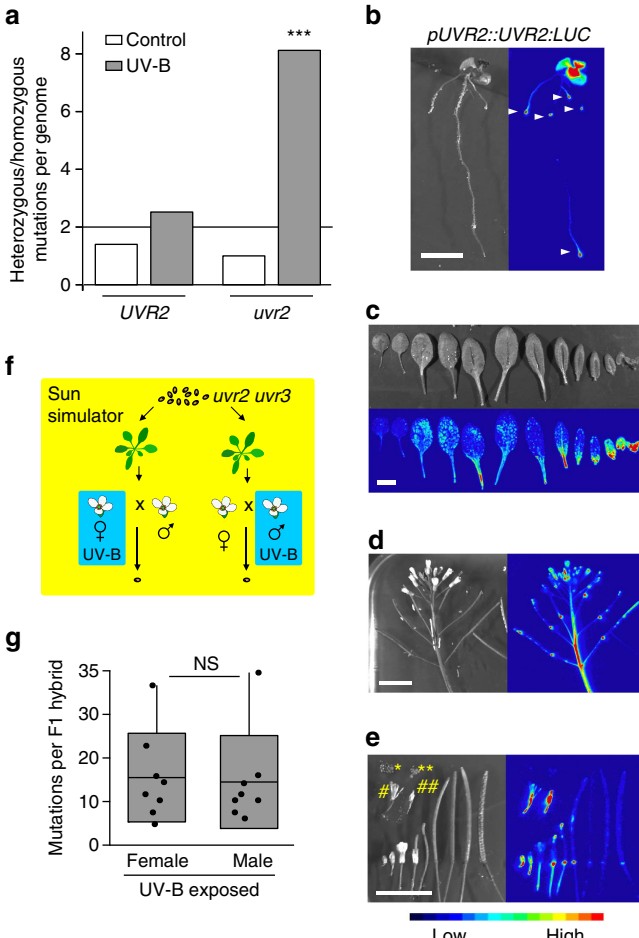

**Figure 3 | Developmental aspects of mutagenesis by UV-B treatment.** (**a**) Ratio of heterozygous versus homozygous mutations in UVR2 (wild-type, *uvr8*, *tt4* and *uvr3*) and *uvr2* (*uvr2* and *uvr2 uvr3*) genotypes after one generation of control and UV-B treatment (300 mW m$^{-2}$ UV-B$_{BE}$). The 2:1 ratio (horizontal line) was expected if all inherited mutations occurred during somatic development. Mutations above this ratio were likely to originate after separation of male and female cell lineages. *** indicates statistically significant differences to all other samples in Fisher's exact test, $P<0.001$. (**b–e**) Expression of UVR2-LUCIFERASE translational fusion construct driven by endogenous promoter (*UVR2promoter::UVR2:LUCIFERASE*). Images on the top/left show plant tissues under white light and those on the bottom/right luciferase signal. All luciferase images were taken using identical exposure time of 1 min, and colour scale at the bottom indicates signal intensity. (**b**) Ten-day-old *in vitro* grown plant. Arrowheads indicate luciferase signals in root apical meristems. Scale bar, 5 mm. (**c**) Leaves dissected from 3-week-old *A. thaliana* plant organized from the oldest (left) to the youngest (right). Scale bar, 10 mm. (**d**) Inflorescence. Scale bar, 10 mm. (**e**) Flower, silique and seed developmental series. Bottom row, left to right: closed flower, flower with emerging pistil, fully opened flower, siliques at different stages and the last opened silique containing seeds with mature embryos. Hashes: pistils and anthers from (#) opened and (##) closed flowers. Petals and sepals were manually removed. Asterisks: (*) dry and (**) fresh seeds. Scale bar, 10 mm. (**f**) Genetic test for sex specificity of UV-B-induced mutations. *uvr2 uvr3* control and UV-B-irradiated plants (300 mW m$^{-2}$ UV-B$_{BE}$) were reciprocally crossed and the number of female- and male-specific mutations was analysed in progeny plants. (**g**) Boxes show genotype average (middle line), s.d. (left and right margins) and values outside of the s.d. range (horizontal bars) between eight analysed genomes (dots) per experimental point. NS, not significant (Student's $t$-test, $P = 0.844$).

direction were sequenced and analysed. All recovered mutations were heterozygous, excluding self-pollination in any of the 16 analysed genomes (Supplementary Data 2). We found on average 12.4 mutations per UV-B-irradiated mother and 13.3 per UV-B-irradiated father, respectively (nonsignificant in Student's $t$-test, $P = 0.844$; Fig. 3g and Supplementary Table 2). This suggests that UVR2 is required for protection of both female and male genome stability, and UV-B treatment induces a similar number of mutations in both sexual lineages.

## Discussion

Land plants are exposed to solar UV-B during their entire life[3]. In order to minimize UV-B-induced damage, plants use multiple protection and repair pathways, including flavonoid sunscreen, direct reversal of pyrimidine dimers and NER[6,8,15,29,30]. We determined the frequency of transgenerationally inherited mutations induced by UV-B treatment in *A. thaliana* wild-type and mutant plants treated with simulated solar UV-B, resembling natural conditions from Helsinki (south Scandinavia) to Madrid (central Spain).

The simulated natural UV-B conditions had only a minimal effect on the rosette growth of wild-type Col-0, indicating that they were well in the photomorphogenic range. A wild-type-like phenotype of the UV-B photoreceptor mutant was unexpected as *uvr8* was found to be UV-B-hypersensitive in previous studies[19,31,32]. The most likely reasons were acute UV-B stress doses applied to non-acclimated plants and/or use of mutants in more sensitive genetic background in the other studies. In contrast, *tt4* and *uvr2* plants were highly sensitive to the simulated natural UV-B, suggesting that flavonoid production and CPD repair, respectively[6,13], are the most important mechanisms sustaining plant growth under simulated natural UV-B.

Under control conditions, we observed on average $2.3 \times 10^{-8}$ mutations per site, which is approximately threefold more than the previously estimated mutation rates of $7.1–7.4 \times 10^{-9}$ for *A. thaliana*[22,33]. This could be because of presence of UV-A and/or higher photosynthetically active radiation (PAR; 400–700 nm; 340 µmol m$^{-2}$ s$^{-1}$) fluence rate applied in our control treatment compared with a typical *A. thaliana* growth chamber (100–150 µmol m$^{-2}$ s$^{-1}$). However, PAR applied in this study corresponds to a partially shaded natural site, while the full exposure to the sun is simulated using much higher PAR fluence rates (800 µmol m$^{-2}$ s$^{-1}$; refs 11,19). Simulated natural UV-B conditions caused only small (1.2–2.2-fold) increase in mutation rates of Col-0 wild-type plants. This is in agreement with a previous study, where simulated solar UV-B regimes provoked only one to four germinal somatic homologous recombination events per 250,000 seedlings[11].

The robust protection of *A. thaliana* transgenerational genome stability against UV-B strongly depends on direct reversal by UVR2 CPD photolyase (summarized as schematic model in Fig. 4). The *uvr2* plants accumulated, on average, 64.3 new mutations per haploid genome and generation under the simulated central Spain UV-B regime. Some of these mutations apparently led to a loss of function for housekeeping genes within just three generations. In contrast, loss of *UVR3* and *UVH1* resulted in a significant, but much lower number of mutations. This may reflect low abundance of UV-B-induced (6–4)PPs (10–25%) relative to CPDs (75–90%) and partial redundancy of NER and UVR3 in repair of (6–4)PPs but not CPDs in *A. thaliana*[13,29].

DNA sequences prone to accumulate UV-B-induced mutations have been unknown in plants. We showed here that sensitivity to our UV-B treatment is determined by both genetic and epigenetic means. Mutations occurred preferentially in the TC dipyrimidine sequence context, and were enriched at methylated cytosines. This differed from spontaneous mutations, which were determined mainly epigenetically by DNA-methylated sites in transposons, but showed no association with particular short sequence motifs. The typical *A. thaliana*-hypermutable sequence TC(C/T) identified here differed from those in humans in at least two aspects. First, we did not observe any CC to TT dinucleotide mutations, which were found frequently in the human eyelid cells[34]. Second, in human skin cells the mutated cytosine was frequently followed by a guanine ((T/C)CG)[23]. A high proportion of (T/C)CG mutations in humans is most likely caused by the enhanced formation of pyrimidine dimers at methylated cytosines[23,24,35,36], which are found exclusively in the CG context in mammalian somatic cells[37]. Absence of such pattern in *A. thaliana* can be explained by presence of DNA methylation in any cytosine context in plants and low number of methylated cytosines in the *A. thaliana* genome[25,26]. Although mutations induced by our UV-B treatment were enriched in *A. thaliana* at the positions of methyl-cytosines (27%) relative to genome background (15%), they were not limited to them, and majority of the mutations (73%) appeared at non-methylated positions. This trend was weaker for spontaneous mutations (60% at non-methylated sites) and suggested that UV-B and spontaneous mutations may quantitatively differ in generating C→T transitions via indirect (involving uracil intermediate) or direct conversion, respectively[38].

Animal male and female germline cells separate from somatic cell lineages early during embryo development, and the latter do not divide any more during the post-embryonic phase[39]. In contrast, plant germline cells with undifferentiated sex divide several times during vegetative growth and separate into male- and female-specific cell lineages only during late flower development[40]. This potentially increases the risk of

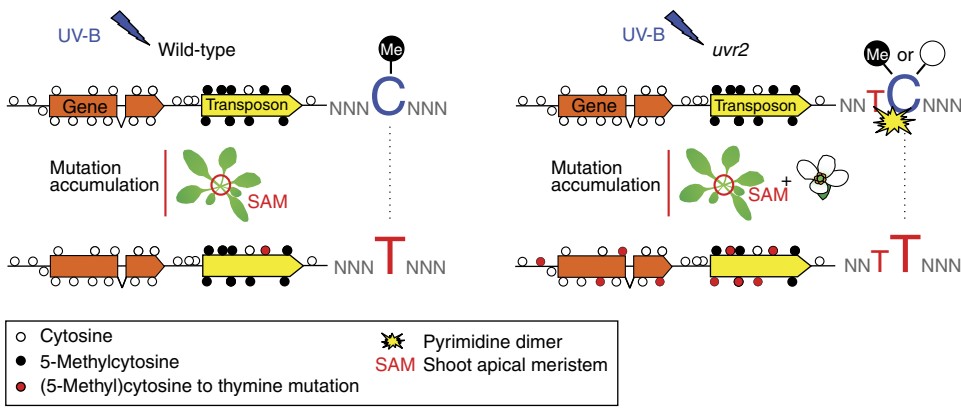

**Figure 4 | Model for accumulation of UV-B-induced germline mutations in *A. thaliana*.**

inheriting mutations via somaclonal sectors. In the first post-irradiated generation of control and UV-B-irradiated plants, we found ∼1:2 ratios of homozygous and heterozygous mutations, respectively. This showed that the spontaneous mutations occurred before the split of male and female cell lineages and the same was true also for mutations induced by UV-B treatment in *UVR2* plants. However, there were fourfold more heterozygous mutations in progenies of UV-B-irradiated *uvr2* plants. This provided strong genetic evidence that UVR2 prevents UV-B-induced mutations in germline cells mainly after separation of male and female cell lineages, and this UVR2 function seems complementary to its role in resolving CPDs in somatic cells[13]. In mammals, mutation rates can be much higher in male than in female gametes[39]. Here we showed that *uvr2* plants derived from UV-B-irradiated male and female reproductive tissues carry almost identical numbers of mutations, suggesting that male and female mutation rates may be more equal in plants. Mammalian mutation bias is caused by accumulation of mutations from DNA replication errors in sperms, which are products of many more cell generations than eggs[39]. It is unknown how many cell divisions (and DNA replications) are required for the development of *A. thaliana* anthers and carpels; however, the information is available from meiosis onwards. At the onset of meiosis there is a single round of DNA replication followed by two rounds of cell division. Subsequently, the released microspore undergoes two rounds of DNA replication and cell division resulting in one vegetative and two sperm cells. The megaspore replicates and divides three times and produces embryo sac with seven nuclei, including haploid egg cell[41]. Hence, there is comparable number of DNA replications in plant mega- versus microgametogenesis. This may explain similar number of mutations observed in our experiments; however, on the other hand it also shows that CPD direct reversal is important in both *A. thaliana* sexual lineages. This is unexpected because eggs are embedded much more in plant tissues than pollen and, therefore, should receive less UV-B damage. We speculate that this may be due to greatly reduced (haploid and unreplicated) genome constitution during gametogenesis, which may limit availability of homologous chromosomes and sister chromatids for homology-based DNA damage repair.

In addition to its activity in somatic cells, direct reversal of CPDs by UVR2 is the key mechanism protecting integrity of DNA from UV-B-induced mutations in *A. thaliana* male and female germline tissues. Direct reversal activity may be particularly important during plant haploid stage, when homology-based repair pathways may not be fully effective because of limited template availability. Therefore, UVR2 is necessary to avoid solar UV-B-induced genetic defects that could be transmitted to the future generations.

## Methods

**Simulation of solar radiation.** Simulation of solar radiation was performed in the sun simulators of the Research Unit Environmental Simulation at the Helmholtz Zentrum München, Neuherberg, Germany. Simulated spectra (280–850 nm; Fig. 1a and Supplementary Fig. 1a,b) were obtained by a combination of metal halide lamps (HQI/D, 400 W; Osram, München, Germany), quartz halogen lamps (Halostar, 300 and 500 W; Osram), blue fluorescent (TLD 18, 36 W, Philips, Amsterdam, the Netherlands) and UV-B fluorescent tubes (TL12, 40 W, Philips). The natural balance from ultraviolet to infrared radiation was achieved by filtering through borosilicate, lime and acrylic glass filters and a water layer and measured using a double monochromator system (Bentham, UK). The filtering in control condition excluded the entire UV-B, present in UV-B treatments. Owing to filter characteristics, ∼80% and more of UV-A were transmitted at control conditions for wavelength >360 nm compared with UV-B treatments, whereas at shorter wavelength of 330 nm only 10% were transmitted (Supplementary Fig. 1a,b). The standard growth conditions were set to resemble the main *A. thaliana*-growing season: day = 14 h, 21 °C, relative humidity 60%, PAR = 340 μmol m$^{-2}$ s$^{-1}$, which resembles natural PAR at shady sites; night = 10 h, 16 °C, relative humidity 80%, no PAR, UV-B radiation 1 h after onset of PAR for 10 h. Dusk and dawn was simulated by switching on/off different

groups of lamps. Four irradiation conditions were applied corresponding to: 0 (control), 100, 150 and 300 mW m$^{-2}$ UV-B$_{BE}$ normalized at 300 nm according to the generalized plant action spectrum[21] (Fig. 1a and Supplementary Fig. 1b). This realistically mimics UV-B$_{BE}$ doses during spring in northern mid-latitudes (40°N, 50°N, 60°N) at, for example, Madrid, Berlin and Helsinki, respectively. The simulated UV-B$_{BE}$ (ref. 21) dose of 300 mW m$^{-2}$ (ultraviolet index = 6; UV-B = 1.2 W m$^{-2}$), applied widely in this study, matched well the integrated values of the spectral irradiance in Madrid (UV-B$_{BE}$ (ref. 21) = 265 mW m$^{-2}$; ultraviolet index = 7; UV-B = 1.3 W m$^{-2}$; modelled for 30 March 2015, 12:00 GMT (total ozone column of 300 DU, surface albedo of 0.1), using the Tropospheric Ultraviolet and Visible model; http://cprm.acom.ucar.edu/Models/TUV/Interactive_TUV/; Fig. 1a).

**Plant material.** Following *A. thaliana* homozygous genotypes in Col-0 background were used: wild-type; *uvr8-6* null[19] (SALK_033468), *tt4* (SALK_020583C), *uvh1* (SALK_096156C), *uvr2* (WiscDsLox466C12), *uvr3* (WiscDsLox334H05) and *uvr2 uvr3*. Each genotype was amplified twice by a single seed descent to reduce any potential heterozygosity, and the resulting seed population was bulk-genotyped before mutation accumulation experiments (Supplementary Fig. 1c). Seeds were sown on a standard soil, and 15 plants per genotype were kept in the described UV-B conditions until seed harvest. Using a single seed descent amplification strategy, we produced three UV-B-irradiated generations (Supplementary Fig. 1c). Note that the sequenced and the irradiated plants were not identical, but siblings (that is, seeds from G1 UV-B-irradiated patent were split into several parts. One part was grown in sun simulator as UV-B-irradiated G2 and the second part was grown in non-UV-B chamber to obtain material for sequencing). This was done in order to avoid stressing UV-B-irradiated plants by additional wounding damage that could potentially influence mutation frequencies.

The *UVR2promoter::UVR2:LUCIFERASE* reporter line was constructed using the Gateway System (Invitrogen) and the Gateway binary vector pGWB435 was used to fuse firefly's *LUCIFERASE* gene to the C terminus of UVR2. The line was stably expressing the construct over multiple generations and T-DNA was excluded to disrupt a gene open reading frame by mapping T-DNA position using TAIL-PCR.

**Nucleic acid isolation and whole-genome sequencing.** From 15 irradiated plants per generation, genotype and treatment, we selected randomly five individuals and grew one progeny plant per individual in a chamber without UV-B radiation for 3 weeks. Subsequently, vegetative rosettes were harvested and DNA extracted with a Nucleon Phytopure Kit (GE Healthcare). Sequencing libraries were prepared using a TruSeq DNA Kit (Illumina). Fragment sizes and library concentrations were assessed on a Bioanalyzer (Agilent) and high-quality libraries were 100 bp paired-end-sequenced on a HiSeq2500 (Illumina) instrument to an average 35× genome coverage (Supplementary Fig. 2 and Supplementary Data 1).

**Mutation detection and validation.** Reads were adaptor- and quality-trimmed using SHORE (v8; ref. 42). Filtered and trimmed reads where aligned to Col-0 reference sequence (TAIR10, 119 Mbp) using GenomeMapper[43] integrated in SHORE (v8) using a maximum of 5% of the read length as mismatches including a maximum of 5% gaps. Read pair information was used to help to remove redundant alignments. Only uniquely mapped reads (after read pair correction) were considered. In order to remove reads originating from the same molecule (because of PCR amplification), we also removed reads with identical 5′ alignments using SHORE. Next, we generated a genome matrix containing information on total coverage and the single base counts for A,C,G,T,- and N for each re-sequenced genome at each reference sequence position. Positions covered by <20 reads were marked as low coverage. All other positions were classified as follows: (i) homozygous wild-type, (ii) homozygous mutant, (iii) heterozygous or (iv) undefined based on the allele frequency of the non-reference alleles. Frequency thresholds were determined empirically (Supplementary Data 1 and Supplementary Figs 6 and 7). Low complexity and tandem repetitive genome regions (comprising 2.95 Mb of the reference sequence), identified by RepeatMasker and TandemRepeatFinder, were excluded during this step to avoid false-positive mutation calls.

Novel mutations should be specific to the genome under consideration (focal genome). Therefore, we compared the variant/allele call in the focal genome with the alleles in nine other genomes of the same genotype (using only the first generation). For focal genomes in generations two and three, we excluded the respective parental genome from this filtering step. A variant call was considered a novel mutation, if none of the other nine genomes showed the same variant and at least six of them showed evidence for a homozygous wild-type allele at this position (Table 1). In addition, we used the following criteria for background filtering: (i) more than one of the background genomes is labelled 'undefined'; (ii) one of the background genomes shows a different homozygous or heterozygous mutation at the same base position; (iii) more than three of the background genomes are insufficiently (<20×) covered; or (iv) less than six background genomes have homozygous wild-type allele calls at the respective position.

We kept track of each position that could be analysed in the focal sample even if the position was called homozygous wild-type (accessible sites), in order to assess the frequency of mutated versus non-mutated accessible sites. The accessible sites

### Table 1 | Mutation classification thresholds.

| Frequency | Classification |
|---|---|
| >0.9 | Homozygous mutation, accepted |
| 0.8–0.9 | Undefined, mutation not accepted |
| 0.3–0.8 | Heterozygous mutation, accepted |
| 0.1–0.3 | Putative sequencing error, not accepted |
| <0.1 | Reference allele, accepted |

included ~75% of the ~120 million sites of the nuclear genomes. Normalized number of mutations per genome was calculated as $n$, where: $n = ((\text{total genome}/\text{accessible genome}) \times \text{number of accepted mutations})/\text{number of treated generations}$. Assignment of mutations to different genome regions (genes, TEs and intergenic regions) was carried out using current *A. thaliana* genome annotations (TAIR10) for genes and TEs. If a TE overlapped with a gene model, we considered the overlapping part as TE, based on the notion that this is frequently DNA-methylated in all cytosine contexts. TE genes were also treated as TEs in our analysis.

**Estimation of false mutation rates with simulated data.** We introduced 900 *in silico* mutations into the Col-0 reference sequence (TAIR10); 308 were homozygous and 592 were heterozygous reflecting the spectrum of mutations reported in this study. We simulated 25 Mio 100 bp Illumina read pairs with an insert size of 370 bp and a sequencing error rate of 2% using wgsim (https://github.com/lh3/wgsim). The sequencing depth for the simulated genome was $41\times$, which is even slightly lower than the average coverage obtained for the real data ($60\times$). The analysis was performed as described before, and nine of the sequenced G1 Col-0 (five control and four Madrid-like UV-B) genomes were used for filtering as background genomes.

The allele frequency distribution for variable sites in the simulated genomes was similar to the distributions observed in real data (Supplementary Figs 6 and 7). However, as the simulated data showed many more variable sites, the simulated sequencing error rate (2%) appeared to be higher than in real data. We found a clear separation in allele frequencies of homozygous and heterozygous variants (Supplementary Figs 6 and 7b). However, the distribution revealed that many of the putative heterozygous variants with an allele frequency between 0.1 and 0.2 are masked by a huge amount of putatively erroneous sites with low mutant allele frequencies. In contrast, only a much smaller number of putative heterozygous sites was observed with an allele frequency between 0.2 and 0.8 in both data sets (Supplementary Figs 6 and 7a). Assuming that the frequencies of real heterozygous sites should be normally distributed with a mean of 0.5 implies that variants with a frequency <0.3 seemingly include a lot of false-positives. The minimal turning point at 0.3 in histogram indicates that using this as a cutoff ensures that we exclude the majority of false-positives while sacrificing only a very small number of true-positives. We found in total 91,500,586 (75% of the genome) accessible sites in the simulated data, which is similar to the real data. In all, 24% of the simulated mutations were in regions that were not accessible according to our definitions. Note that this does not affect the mutation rate estimations as mutation frequency is estimated across the number of accessible sites. Of the remaining 685 *in silico* mutations located at the accessible site, 684 were identified by our approach (Supplementary Fig. 7b). Only one heterozygous mutation could not be reported, as it had an allele frequency below 0.3. Together, this simulation suggests a false-negative rate of 0.15%. We did not encounter any false-positive in this simulation, suggesting that our strict cutoffs are very robust against false-positives even at high sequencing error rates. In order to support this finding, we tested a random set of 59 candidates from a total of 2,497 mutations identified in the real sequencing by Sanger sequencing. We were able to confirm 58 of them (Supplementary Data 2).

**DNA sequence motif analysis.** For each accepted mutation, we extracted positions three bases up- and downstream from the respective position. Mutations were grouped by the type of base change (for example, C→T) and the extracted sequences were used as input for the software weblogo v3.4 (ref. 44), which generates bit scores for each base (A, C, G or T) at a specific position. If a base is found more often than expected according to the background probability of each base (here C = G = 0.2, A = T = 0.3), it gets a higher bit score.

**DNA methylation analysis.** DNA methylation data were retrieved from publicly available wild-type *A. thaliana* data sets GSM980986, GSM980987 and GSM938370 (ref. 26). Only nucleotide positions with ≥10 sequencing reads were considered for analysis. A cytosine was considered as methylated if its methylation frequency reached ≥10% in at least two biological replicates. Because these criteria are partially different from those applied in other studies[25,26], we obtained generally higher DNA methylation frequencies. Statistical significance of the results was tested as the number of methylated and unmethylated cytosines in sample A versus sample B using Chi-square test with Yates correction.

**Data availability.** Illumina reads generated in this study have been deposited to the European Bioinformatics Institute (EBI) database under the accession numbers (PRJEB13889; http://www.ebi.ac.uk/ena/data/view/PRJEB13889). All other data supporting the findings of this study are included in the manuscript and its supplementary files or are available from the corresponding authors upon request.

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

## Acknowledgements

We thank Chris Bowler and Jean Molinier for *uvr2*, *uvr3* and *uvr2 uvr3* seeds, Holger Puchta for *uvh1* seeds, Roman Ulm for *uvr8* seeds and Hume Stroud and Matteo Pellegrini for bisulfite-sequencing raw reads. We are grateful to Regina Gentges, Barbara Eilts, Petra Pecinkova (all MPIPZ), S. Mühlhans and T. Gartner (both HMGU) for technical assistance, and Maarten Koornneef, George Coupland, Jörg-Peter Schnitzler and Ortrun Mittelsten Scheid for critical reading of the manuscript. This work was supported by general funds of the Max Planck Society to A.P. and K.S. and COST action FA0906 (UV4Growth) to A.P., T.P., A.A. and J.B.W.

## Author contributions

A.P., T.P. and K.S. designed the experiments. A.A. and J.B.W. planned and performed sun simulator experiments and determined growth parameters. T.P., A.A. and J.B.W. grew the plants. T.P. prepared sequencing libraries. E.-M.W. analysed whole-genome sequencing data and identified mutations. T.P. and E.-M.W. analysed mutation spectra and associations with genomic features. T.P. cloned the *UVR2* reporter construct. A.P. wrote the manuscript with contribution from all authors. All authors read and approved the submitted manuscript.

## Additional information

**Competing financial interests:** The authors declare no competing financial interests.

