## [Peer Review File · Nature Communications]

Reviewers' comments:

Reviewer #1 (Remarks to the Author):

This is a very nicely designed study that measures the influence of high- but reasonably natural, embedded in a naturalistic simulated solar spectrum- UVB levels on heritable mutation in Arabidopsis. As we know from studies in viruses, bacteria, animals, plants and fungi, UV induces primarily pyrimidine dimers, and cyclobutane pyrimidine dimers make up the majority of those. As we already know from earlier Arabidopsis work, Arabidopsis repairs CPDs mainly through the action of CPD photolyase, with some minor, strand specific transcription coupled repair by NER. (It's a quite misleading that the authors suggest that "little is known about how plants protect against UVB induced mutations- we know exactly which repair pathways are employed against UV induced damage, and we have assumed that in plants, like all other organisms, that failure to repair these lesions would result in enhanced mutagenesis, with a particular signature of C->T at dipyrimidines.) We also assume that the same repair pathways exist in the germline as have been demonstrated to function in the plant overall. However, no one has ever actually measured the frequency and spectrum of UV-induced mutations specifically in plants. The authors do that here. In addition, by looking at the ratio of homozygous vs. UV-induced heterozygous mutations in the progeny of UV-exposed, CPD-repair defective plants, they conclude- correctly- that many of these mutations are induced late in development, after the divergence of the male and female reproductive organ cells types within a floral primordium, in the more light-exposed exposed floral tissues, rather than the relatively UV-protected apical meristems. So far I'm with them, and I think they've done a very nice job. The result isn't surprising, but its well supported. Where the authors and I diverge is in their claims of novelty:

The conclusion of the discussion is- to me at least- ambiguous- what exactly is the a plant-specific adaptation discussed here? That the repair of UV-induced damage by a CPD-specific photolyase is an important pathway for the reversal of potentially CPDs? We know that already, and it isn't plant specific. That repair occurs in the germline? Are the authors really suggesting that this repair occurs only in the germline, and that that's a peculiarity of plants? That's not true- repair of CPDs by photolyase has certainly been demonstrated in other (not male or female germline-specific) Arabidopsis tissues. They also state in the abstract that "plants possess a unique protection against the mutagenic effects of sunlight"- again, photolyase is not unique, they seem to be focusing on the enhanced expression of transcript for photolyase observed in flowers. This pattern of expression- enhanced expression in mitotically active cell populations (and endoreduplicating cells) has been previously observed and is true for the majority DNA metabolism-related genes- including repair genes- in Arabidopsis. Again, this is a well-performed nice study but not a surprising result. It confirms our assumptions. It will certainly be widely cited in plant DNA repair literature.

Here is a list of other suggestions"

Uvr3 and uvh1 are two pathways that both repair 6-4's. Whereas CPDs are rarely repaired via NER (during transcription-coupled repair only). UVR2 does most of the job. I would

consider this before concluding that UVR3 is not important simply because 6-4s are relatively rare.

Before concluding that UVR2 is uniquely expressed in flowers, could we have a look at, for example, seedlings? Even roots? Emerging lateral shoots? Very young leaves? These are very easy experiments, given that the authors have a luciferase fusion. The author might change their minds and conclude that UVR2 is most highly expressed in actively replicating cells.

Could we have a little more information about what was and wasn't sequenced? I'm assuming you sequenced only those reads that mapped uniquely? Is the discussion of TEs then actually a special subset of TEs that are present only in single copy, or fragments of TEs that carry unique sequences adjacent to their borders? Are the transposons transcribed or not? Could transcription-coupled repair affect the mutation rate?

P8 bottom should be "in the progeny of the 1st generation of plants exposed to UVB"?

P9 top- Its difficult to distinguish between germline and somatic cells in plants- to clarify your point and to help out any non-plant biologists, how about something like: "Arabidopsis flowers produce both pollen and ova in each flower. If all mutations occurred between before the differentiation of the male and female organs, we'd expect a 2:1 ratio..."

P10 middle paragraph: Fix English, it not so good.

P10 "this could be due to the mutagenic effects of UVA-" is this consistent with the mutation spectrum?

The term "control-treatment" used throughout is a little confusing, you might replace it with "control"

P11: "Some of these mutations apparently led a loss of functions from housekeeping genes within just three generations" fix the English and I must have missed the data supporting this statement- just point it out to me.

"contrasted with high mutation accumulation... in humans with dysfunctional NER". This would be a really nice place to point out that humans and other placental mammals are almost unique among all living things in that they lack photolyases. A more sophisticated discussion of the unique importance of UVR2 (CPD photolyase) vs. UVR3 (6-4 photolyase) might include the fact that NER. In plants and some animals, repairs very few of the CPDs (it recognizes them only in the context of a stalled RNA polymerase), while in contrast the NER and UVR3 pathways are somewhat redundant in that they both repair global 6-4s.

P4 first paragraph under "results": make it clear here that the entire solar spectrum is adjusted to match the UVB levels for Madrid (a supplementary graph illustrating the actual vs. simulated spectrum would be nice?). ("Methods" states this but tell us in the text too- that's a very impressive facility.

P5 l108 states that the effects of UVB on mutation frequency in wt in this very high (though natural) UVB environment are statistically insignificant. Yet this doesn't stop the authors from stating that "simulated UVB... increased mutation rates 1.2 to 2.2 fold" and concluding that "solar UVB may increase mutation rates by several fold in Arabidopsis natural populations". Yes, this is possible, but your data doesn't support any particular conclusion. I would recommend that this paragraph be dropped. Or replaced with one that simply states that, at the statistical power provided by this study, there is no evidence to indicate that even very high natural UVB flux increases germinal mutation rate in wild-type plants.

P12 291 "UVR2 transcript accumulation may reflect major transcriptional reprogramming" Is this worth devoting a paragraph to, especially given that none of the data comes from this paper?

This study might actually be performed under natural light to see if the authors reach the same conclusions, but that could certainly be a later paper.

In summary this is an elegant and well-performed study that quantifies the importance of various repair pathways in protecting the Arabidopsis genome from UV exposure. The study, importantly, employs a solar simulator to invoke the plants' natural response to solar UV. The authors conclude that CPD photolyase is especially important in preventing germline mutations. This result is not surprising given that CPDs make up the majority of UV induced DNA damage and that the vast majority of their repair proceeds via this enzyme. I do disagree with some of the authors claims regarding the novelty of this mechanism for the prevention of mutation.

Reviewer #2 (Remarks to the Author):

DNA damage caused by ultraviolet radiation has been widely accepted as an important source of both somatic and germline mutations. This paper evaluated the effect of UV-B on germline mutations of Arabidopsis wild type and 6 mutants by genome sequencing. In addition, this study investigated the effects of sequence specificity, DNA methylation, UV responsive genes and somatic mutation inheritance. They find that "the plant-specific UVR2 photolyase is the major mechanism required for sustaining plant genome stability across generations", based on the largely increased mutation rates in *uvr2* mutant. This is a systematical mutation study by directly sequencing many mutants and wild Col at various doses of UV light which provides novel insight and be of interest to others in this field. However, I still have some concerns.

1. As the major conclusion (UVR2 is the major mechanism for genome stability under UV), it is better to exclude the other possibility (e.g., the other genes). In this study, no solid evidence could exclude the gene UVR-8. SALK_033468 is used as *uvr-8* mutant in this study, and this mutant only has minor increase in mutation rates. Actually, this result may be not surprising because SALK_033468 only has a T-DNA insert in the intron of UVR-8,

according to description of TAIR. Intronic T-DNA insertion could also have intact transcripts, so perhaps this [j1] line is not a null mutant of UVR-8. Maybe a null line could be found from more T-DNA libraries, CRISPR-knockout lines or from the original reference which showed that UVR-8 is a trigger of UV response.

2. Mutation numbers could be quite different among individual plants even within the same lineage. However, some control groups only have four samples, which could produce the biased results. Maybe more samples are necessary.

3. As I know, the mutation numbers observed are very sensitive to the criteria of identification because mutation is rare in a genome. Detecting mutation based on those given frequencies seemed a bit arbitrary. The frequency is subjected to uneven depths in different nucleotide positions due to sequencing artifacts or mapping errors. Assuming a position with 60 reads, 10 from PCR duplicates, 15 support for true mutations, a 0.1-0.3 criteria would reject such a mutation while the true frequency should be $15 / (60-10) = 0.3$. Therefore, an assessment of the false negative rate is necessary, which was not mentioned in this study.

4. For the verification of mutations, perhaps a more detailed description is desirable. Were the mutations only confirmed in the samples supposed to carry this mutation? It is better to select the other samples, especially their parents, to verify as controls. As the number of mutations was quite different from those reported before, it's better to make every effort to make every methods clear for readers.

5. Given that in this study, many samples have been cultivated for 3 generations, have the inheritance of mutations in progenies been tested? Segregation ratios of homozygous and heterozygous mutations can also be checked.

6. At line 178-184, different regions with different methylation patterns have different mutation numbers. How much could this difference be explained by different Cytosine mutation rate?

7. C->T mutation increase is a major signature of UV radiation. In 2015, a systematic study of eyelid sequencing has found that CC->TT dinucleotides mutation would also increase under UV exposure (Martincorena, I. et al, 2015) .Could this be observed in this study?

1) Martincorena, I. et al. High burden and pervasive positive selection of somatic mutations in normal human skin. Science 348, 880-886 (2015).

Reviewer #3 (Remarks to the Author):

A-B. Summary of key results and originality or interest.

I believe the most novel and important result is the finding that UVR2 protects from UV-induced mutations especially in the germline after the split between male and female tissues. And that both gametes contribute a comparable amount of UV-induced mutations to the next generation. I think these are unexpected, well supported findings that will inspire future work and theory development. They are accompanied by many other results that may not make the same impact, because they pretty much confirm the expectations. For example, the relative importance of the pathways affected in UV-sensitive mutants is important, but does not pose further exciting questions by itself. In my opinion, the same is true about the spectrum and distribution of UV-induced mutations along the genome.

C. Data & methodology

The dataset is excellent and I think the methods are appropriate. My only disappointment was the low resolution of the Luciferase-tagged UVR2 expression. To see the flower lightened in Figure 3b is compatible with the idea that most UV-induced mutations prevented by UVR2 would have happened in the germline after the split of male and female organs. I hoped that a closer look would have shown what parts of the flower express more UVR2. However, a higher resolution is not essential to hold the main result.

D. Use of statistics

I have only two minor concerns about statistics. One is the general use of Fisher's exact test. This test is used to check the independence between two categorical variables. Thus, I think it is appropriate to see if mutations appear more or less frequently than expected in methylated or non-methylated cytosines, for example. In several other cases, I agree it's well used. However, I cannot understand how Fisher's exact test is used to compare the normalized number of mutations per genome between two treatments or two genotypes.

My other request is to see, when possible, the p-value, rather than just saying it's significant at 0.05 level.

E. Conclusions.

The reasoning behind the main conclusions is sound. From the 8:1 ratio of heterozygous to homozygous mutations in UV-treated *uvr2* plants is natural to conclude that most mutations happen after the split between female and male cell lineages. The mating experiment and the conclusions from it are also correct.

F. Suggested improvements.

I encourage the authors to elaborate the implications of the main conclusions. How does it

relate to what is known about the development of the flower? When do the male and female cell lineages split? The similar numbers of mutations in male and female cell lineages requires a better explanation than a "similar number of cell cycles leading to plant male and female gametes compared to animals". It is evident to the broad audience that there are many more pollen cells than ovules, suggesting a very different number of cell cycles.

If proper importance is given to these issues, some re-structuring of the text could improve readability and interest. It is unfortunate to find these nice results hidden behind lengthy descriptions of mutation counts and non-significant or mildly interesting comparisons. It may be possible to shorten it.

Some details:

- Supplementary figure 4 is actually a table.
- On supplementary table 1, it seems that the mutations per site "per generation" are not given "per generation" when more than 1 generation, but the accumulated after 2 or 3 generations. Just for clarity.

G. References.

Honestly, I didn't check if there are some better references.

H. Clarity and context.

I don't think the abstract makes justice to the paper. It presents it as if we didn't know much about how plants protect themselves from UV radiation. But the genes involved are already described. The real contribution is the precise measurements of the numbers of mutations prevented by these pathways in different conditions, which provides theoreticians with evidence to test hypotheses about the evolution of plants and pathways.

Reviewer #4 (Remarks to the Author):

This is an interesting paper with some potentially important conclusions. However, there are a number of serious concerns about the data and the validity of some of the analyses and conclusions.

Major Concerns:

1. The UV-B exposure experiments do not use an appropriate control. The UV treatments are additionally exposing the plants to high levels of UV-A (~75% of total UV) as well as to UV-B at different levels, but this UV-A is not present in the control treatments (Supp. Fig. 1A). The effects observed in the exposed lineages could thus be caused by this UV-A, rather than by the UV-B which the paper is supposed to be about. The experiments need a proper control (as for example is in the cited Ries et al., 2000 paper). Without this kind of control it

is impossible to determine how much of what is seen is genuinely due to UV-B exposure.

2. The genome coverage ('accepted positions') is poor, in some cases less than 60% of the genome. This is well below the standard accepted for this kind of analysis (usually 90% or above), and means that many of the conclusions reached in the manuscript may not be robust. The samples need to be sequenced to better coverage.

3. The title is inaccurate. Natural UV-B is not studied in this work. '...trans-generational genome stability under simulated natural UV-B....' (or something similar) would have been better.

4. The Abstract claims that '...UVR2 photolyase is the major mechanism required for sustaining plant genome stability across generations'. This 'major mechanism' conclusion cannot be drawn from the work described in this manuscript. The only conclusion that can potentially be drawn (provided the observations are shown to be robust) is that UVR2 provides a mechanism for protecting against UV damage. Other mechanisms (e.g., mismatch repair) are likely as (if not more) important in 'sustaining genome stability across generations'. The authors need to be more circumspect in the accuracy of their writing (a criticism that applies to throughout the entire manuscript, and not just in this specific instance).

Additional concerns/comments:

1. Pg. 4 and Fig. 1b. The growth phenotypes shown are interesting. It makes sense that the *tt4* mutation reduces photoprotection and hence growth in increased UV. But why should lack of UVR2 (a DNA repair enzyme) in *uvr2* affect growth in increased UV? It is unlikely that this effect is due to somatic mutation as all such mutations would be heterozygous. Perhaps UVR2 has functions other than DNA repair? Could the authors comment?

2. Pg. 5. A false positive discovery estimate is provided. But we also need to know the false negative discovery rate.

3. Pg. 5. The authors report a spontaneous mutation rate which (in Discussion) they claim to be 'higher' than previous estimates. Is this difference statistically significant? If not, don't comment on it and don't say that it is higher. Actually this observation is based on a very small sample size (9, 13 and 10 mutations (Supp. Table 1) is not enough to draw meaningful spontaneous mutation rates), and suspect for that reason. But if the rate is genuinely higher, could it be due to the very high irradiance that the lineages are exposed to (340 $\mu\text{mol m}^{-2} \text{s}^{-1}$ is stressfully high; standard growth conditions are 100-150). Could this be a reason why, if indeed the spontaneous mutation rates are indeed robustly higher, they are 4-fold higher than those observed in the Ossowski et al. paper?

4. Pg. 5. '...1.2-2.2-fold higher than the average number of mutations in control-treated plants, but not significantly higher than expected (Fisher's exact test, $\alpha = 0.05$) given the number of analysed genomes.' If it isn't significant it isn't higher and the authors shouldn't say that it is. This lack of statistical robustness is a thread running through the entire manuscript.

5. Pg. 6. The authors suggest that UV-B-treatment of *uvr2* plants increases the frequency of non-synonymous mutations in protein coding regions. However it is not clear if this increase is simply the consequence of increased numbers of mutations throughout the genome or if the authors are claiming that mutations under these conditions are somehow targeted to protein coding regions. Clearer writing is needed here.

6. Pg. 7. The authors claim that transposons are prone to G:C to A:T transitions under UV-

B-free conditions and that genes are not. This is a very surprising finding. Why was it not found during the analyses described in the Ossowski et al. paper? Have the authors re-analysed the data in the Ossowski et al. paper (they should, because there are more mutations there than in the data in the paper currently under review) to see if they find the same thing? Perhaps this is another case of over-interpreting from insufficient data (insufficient mutations)?

7. Fig. 1b, c. Why draw the histograms in this complicated way (the boxes with individual data points rather strangely arranged). Conventional means plus/minus SE would show all that you need to show.

8. Fig. 2a. What is frequency (what are the units)? Also it is not clear what we are looking at here. It needs to be absolutely clear that this is not just data from WT plants but is actually data from WT plus mutant lines carrying UVR2 (but not homozygous for *uvr2*).

9. Fig. 3a. 'Homozygous/heterozygous mutation...' in the column should be corrected as 'Heterozygous/homozygous mutation...'

Reviewers' comments:

Reviewer #1 (Remarks to the Author):

This is a very nicely designed study that measures the influence of high- but reasonably natural, embedded in a naturalistic simulated solar spectrum- UVB levels on heritable mutation in Arabidopsis. As we know from studies in viruses, bacteria, animals, plants and fungi, UV induces primarily pyrimidine dimers, and cyclobutane pyrimidine dimers make up the majority of those. As we already know from earlier Arabidopsis work, arabidopsis repairs CPDs mainly through the action of CPD photolyase, with some minor, strand specific transcription coupled repair by NER. (It's a quite misleading that the authors suggest that "little is known about how plants protect against UVB induced mutations- we know exactly which repair pathways are employed against UV induced damage, and we have assumed that in plants, like all other organisms, that failure to repair these lesions would result in enhanced mutagenesis, with a particular signature of C->T at dipyrimidines.) We also assume that the same

repair pathways exist in the germline as have been demonstrated to function in the plant overall. However, no one has ever actually measured the frequency and spectrum of UV-induced mutations specifically in plants. The authors do that here. In addition, by looking at the ratio of homozygous vs. UV-induced heterozygous mutations in the progeny of UV-exposed, CPD-repair defective plants, they conclude- correctly- that many of these mutations are induced late in development, after the divergence of the male and female reproductive organ cells types within a floral primordium, in the more light-exposed exposed floral tissues, rather than the relatively UV-protected apical meristems. So far I'm with them, and I think they've done a very nice job. The result isn't surprising, but its well supported.

Response: We thank the reviewer #1 for these positive comments. We have reformulated the text concerning the depth of knowledge on pyrimidine dimer repair in the abstract.

Where the authors and I diverge is in their claims of novelty: The conclusion of the discussion is- to me at least- ambiguous- what exactly is the a plant-specific adaptation discussed here? That the repair of UV-induced damage by a CPD-specific photolyase is an important pathway for the reversal of potentially CPDs? We know that already, and it isn't plant specific. That repair occurs in the germline? Are the authors really suggesting that this repair occurs only in the germline, and that that's a peculiarity of plants? That's not true- repair of CPDs by photolyase has certainly been demonstrated in other (not male or female germline-specific) Arabidopsis tissues. They also state in the abstract that "plants possess a unique protection against the mutagenic effects of sunlight"- again, photolyase is not unique, they seem to be focusing on the enhanced expression of transcript for photolyase observed in flowers. This pattern of expression- enhanced expression in mitotically active cell populations (and endoreduplicating cells) has been previously observed and is true for the majority DNA metabolism-related genes- including repair genes- in Arabidopsis. Again, this is a well-performed nice study but not a surprising result. It confirms our assumptions. It will certainly be widely cited in plant DNA repair literature.

Response: We agree that UVR2 and UVR3 detoxify pyrimidine dimers in differentiated somatic tissues (e.g. leaves). In the first version of the manuscript we mentioned the corresponding publication (Reference #13), but did not highlight this directly. In the revised version we made this more obvious in the introduction and the discussion. We have now rewritten the text to make the novelty of our work more obvious.

Here is a list of other suggestions"

Uvr3 and uvh1 are two pathways that both repair 6-4's. Whereas CPDs are rarely repaired via NER (during transcription-coupled repair only). UVR2 does most of the job. I would consider this before concluding that UVH3 is not important simply

because 6-4s are relatively rare.

Response: We formulated the text more carefully and included discussion on the preferential repair of 6-4 photoproducts by NER.

Before concluding that UVR2 is uniquely expressed in flowers, could we have a look at, for example, seedlings? Even roots? Emerging lateral shoots? Very young leaves? These are very easy experiments, given that the authors have a luciferase fusion. The author might change their minds and conclude that UVR2 is most highly expressed in actively replicating cells.

Response: We thank for this suggestion. We analyzed expression of our UVR2 reporter line in different tissues. The data are now included as Fig. 3b-e and Supplementary Figure 5 (controls). Indeed, UVR2 is expressed not only in flowers, but in all actively dividing tissues, including root tips. This does not contradict our results concerning mutation accumulation in the germline. We have changed the text accordingly.

Could we have a little more information about what was and wasn't sequenced? I'm assuming you sequenced only those reads that mapped uniquely? Is the discussion of TEs then a actually a special subset of TEs that are present only in single copy, or fragments of TEs that carry unique sequences adjacent to their borders? Are the transposons transcribed or not? Could transcription-coupled repair affect the mutation rate?

Response: We now provide more detailed information on processing of next generation sequencing data in the materials and methods section. Indeed, we used only uniquely mapping read pairs. Looking at the different genomic fractions genes, TEs and intergenic space in the Arabidopsis reference sequence, we cover on average slightly less TE space (16.6% compared to 19.1%) and slightly more intergenic space (33% compared to 30.4%) in our analysis relative to the full genome sequence. However, the genome of Arabidopsis contains only a very small fraction of young (identical) TEs. Only ~5.5% of positions in age annotated TEs are in young (100% identical to consensus) TEs, Willing et al. (2015)). Around 11% of the positions are in TEs \geq 99% identical. We sequenced 200 bp (read1 + read2) from each fragment in the library, therefore we can assume that we are able to cover most of the TE space in Arabidopsis and that our findings are not severely confounded by the slight bias towards "older" TEs. However, it might be true that young TEs are slightly underrepresented in our accessible genome space. Arabidopsis transposons are typically not or only very low transcribed due to strong epigenetic silencing. TE-derived transcripts constitute approximately 1% of the Arabidopsis mRNA populations (Pecinka lab, unpublished data). Therefore, we assume only minor contribution of TCR to repair pyrimidine dimers in TEs relative to genes.

P8 bottom should be "in the progeny of the 1st generation of plants exposed to

UVB"?

Response: Corrected.

P9 top- Its difficult to distinguish between germline and somatic cells in plants- to clarify your point and to help out any non-plant biologists, how about something like: "Arabidopsis flowers produce both pollen and ova in each flower. If all mutations occurred between before the differentiation of the male and female organs, we'd expect a 2:1 ratio..."

Response: We corrected the text on page 9 as suggested.

P10 middle paragraph: Fix English, it not so good.

Response: We re-wrote this paragraph.

P10 "this could be due to the mutagenic effects of UVA-" is this consistent with the mutation spectrum?

Response: To our best knowledge, there is not a single base pair resolution study of UV-A mutagenic effects in plants. The other limitation is poor definition of the light conditions in other plant mutation accumulation studies. Therefore, this question cannot be addressed at the current state of knowledge.

The term "control-treatment" used throughout is a little confusing, you might replace it with "control"

Response: Corrected throughout the manuscript.

P11: "Some of these mutations apparently led a loss of functions from housekeeping genes within just three generations" fix the English and I must have missed the data supporting this statement- just point it out to me. "contrasted with high mutation accumulation... in humans with dysfunctional NER". This would be a really nice place to point out that humans and other placental mammals are almost unique among all living things in that they lack photolyases. A more sophisticated discussion of the unique importance of UVR2 (CPD photolyase) vs. UVR3 (6-4 photolyase) might include the fact that NER. In plants and some animals, repairs very few of the CPDs (it recognizes them only in the context of a stalled RNA polymerase), while in contrast the NER and UVR3 pathways are somewhat redundant in that they both repair global 6-4s.

Response: We have now cross-referenced this more thoroughly in the results section (Fig. 1d and Supplementary Tables 2 and 3; please see also example of the semi-dominant dwarf plant mutant in Supplementary Figure 3c). We included the discussion on partial redundancy of UVR3 and NER pathways at suggested position.

P4 first paragraph under "results": make it clear here that the entire solar spectrum is adjusted to match the UVB levels for Madrid (a supplementary graph illustrating the actual vs. simulated spectrum would be nice?). ("Methods" states this but tell us in the text too- that's a very impressive facility.

Response: We included a modeled UV spectrum of Madrid in Fig. 1a from 30/03/2015, 12:00 GMT, using the Tropospheric Ultraviolet and Visible (TUV) model of Sascha Madronich (ACD, NCAR). Due to the fact that is technically very difficult to simulate the exact solar spectrum, a difference between sun simulator and modeled spectrum is obvious. This has more than one reason, e.g. (i) big glass filters cannot mimic the ozone absorption of the atmosphere in the short wavelength range of UV exactly, (ii) different ozone absorption and surface albedo used in the TUV model compared to reality. However, the overall UV spectrum achieved during our experiments was a very good match of the integrated values of the spectral irradiance: Madrid: UV index = 7, UV-B (biologically effective) = 265 mW m⁻² after Caldwell (1971) normalized at 300 nm, UV-B = 1.3 W m⁻²; Sun Simulator: UV index = 6, UV-B (biologically effective) = 300 mW m⁻², UV-B = 1.2 W m⁻². We added this information also into materials and methods.

P5 I108 states that the effects of UVB on mutation frequency in wt in this very high (though natural) UVB environment are statistically insignificant. Yet this doesn't stop the authors from stating that "simulated UVB... increased mutation rates 1.2 to 2.2 fold" and concluding that "solar UVB may increase mutation rates by several fold in Arabidopsis natural populations". Yes, this is possible, but your data doesn't support any particular conclusion. I would recommend that this paragraph be dropped. Or replaced with one that simply states that, at the statistical power provided by this study, there is no evidence to indicate that even very high natural UVB flux increases germinal mutation rate in wild-type plants.

Response: We made clear that the number of UV-B-induced mutations in wild-type plants was small and removed the speculative statements.

P12 291 "UVR2 transcript accumulation may reflect major transcriptional reprogramming" Is this worth devoting a paragraph to, especially given that none of the data comes from this paper?

Response: We removed large parts of this paragraph and merged it with the previous paragraph.

This study might actually be performed under natural light to see if the authors reach the same conclusions, but that could certainly be a later paper.

Response: We agree that this would be an interesting experiment, and we are considering it for a follow up study.

In summary this is an elegant and well-performed study that quantifies the importance of various repair pathways in protecting the Arabidopsis genome from UV exposure. The study, importantly, employs a solar simulator to invoke the plants' natural response to solar UV. The authors conclude that CPD photolyase is especially important in preventing germline mutations. This result is not surprising given that CPDs make up the majority of UV induced DNA damage and that the vast majority of their repair proceeds via this enzyme. I do disagree with some of the authors claims regarding the novelty of this mechanism for the prevention of mutation.

Response: We thank the reviewer #1 for this very thorough review and for the constructive critics.

Reviewer #2 (Remarks to the Author):

*DNA damage caused by ultraviolet radiation has been widely accepted as an important source of both somatic and germline mutations. This paper evaluated the effect of UV-B on germline mutations of Arabidopsis wild type and 6 mutants by genome sequencing. In addition, this study investigated the effects of sequence specificity, DNA methylation, UV responsive genes and somatic mutation inheritance. They find that "the plant-specific UVR2 photolyase is the major mechanism required for sustaining plant genome stability across generations", based on the largely increased mutation rates in *uvr2* mutant. This is a systematical mutation study by directly sequencing many mutants and wild Col at various doses of UV light which provides novel insight and be of interest to others in this field. However, I still have some concerns.*

*1. As the major conclusion (UVR2 is the major mechanism for genome stability under UV), it is better to exclude the other possibility (e.g., the other genes). In this study, no solid evidence could exclude the gene UVR-8. SALK_033468 is used as *uvr-8* mutant in this study, and this mutant only has minor increase in mutation rates. Actually, this result may be not surprising because SAL_033468 only has a T-DNA insert in the intron of UVR-8, according to description of TAIR. Intronic T-DNA insertion could also have intact transcripts, so perhaps this [j1] line is not a null mutant of UVR-8. Maybe a null line could be found from more T-DNA libraries, CRISPR-knockout lines or from the original reference which showed that UVR-8 is a trigger of UV response.*

Response: We discussed the choice of the *uvr8* allele with Dr. Roman Ulm (University of Geneva), leading scientist in understanding UV-B perception in plants, prior to our experiments. He recommended *uvr8-6* (SALK_033468) as the null allele based on the protein blot experiments performed in his laboratory (see Favory et al., EMBO J, 2009, Fig. 3C). The *uvr8-6* material analyzed in our study

originated from the seeds donated by Ulm lab. We emphasized this in the materials and methods section.

2. Mutation numbers could be quite different among individual plants even within the same lineage. However, some control groups only have four samples, which could produce the biased results. Maybe more samples are necessary.

Response: Our study was designed in order to capture the major and the large size effects. Despite existing variation, the large size effects are clearly obvious with the analyzed number of genomes. Estimating small size effects would require massive sequencing of many more genomes, which is clearly beyond our current possibilities.

3. As I know, the mutation numbers observed are very sensitive to the criteria of identification because mutation is rare in a genome. Detecting mutation based on those given frequencies seemed a bit arbitrary. The frequency is subjected to uneven depths in different nucleotide positions due to sequencing artifacts or mapping errors. Assuming a position with 60 reads, 10 from PCR duplicates, 15 support for true mutations, a 0.1-0.3 criteria would reject such a mutation while the true frequency should be $15 / (60-10) = 0.3$. Therefore, an assessment of the false negative rate is necessary, which was not mentioned in this study.

Response: We now describe the different steps in our analysis in more detail. Indeed, PCR duplicates are known not only to be a source of false negatives, but also of false positives, since errors of the DNA polymerase get amplified and appear as a new heterozygous site with high support. Therefore, PCR duplicates are removed before generating allele frequencies per position in the genome.

4. For the verification of mutations, perhaps a more detailed description is desirable. Were the mutations only confirmed in the samples supposed to carry this mutation? It is better to select the other samples, especially their parents, to verify as controls. As the number of mutations was quite different from those reported before, it's better to make every effort to make every methods clear for readers.

Response: We extended materials section describing mutation validation. Mutations were Sanger sequenced only in the samples in which they were identified. However, as many of the analyzed mutations were heterozygous, both alleles were visible in the trace files of the Sanger sequencing results (e.g. the mutated, new base and the wild-type base). In these cases the wild-type base was compared to the wild-type base / sequence in the reference genome and confirmed in all cases.

5. Given that in this study, many samples have been cultivated for 3 generations, have the inheritance of mutations in progenies been tested? Segregation ratios

of homozygous and heterozygous mutations can also be checked.

Response: Unfortunately, the complex genealogy of our material (Supplementary Figure 1) does not allow addressing this question directly. UV-B irradiation and sequencing were performed on siblings derived from the same parental plant, which are expected to carry overlapping but not identical sets of mutations. We confirmed this by observing some of the mutations from the first generation again in the third generation, but not in the second generation. We have now commented on this in the Materials and methods section.

6. At line 178-184, different regions with different methylation patterns have different mutation numbers. How much could this difference be explained by different Cytosine mutation rate?

Response: Here we are limited by the number of spontaneous (control) mutations overlapping with C or mC in different genomic regions. Therefore, we cannot provide any statistically conclusive estimation here.

7. C->T mutation increase is a major signature of UV radiation. In 2015, a systematic study of eyelid sequencing has found that CC->TT dinucleotides mutation would also increase under UV exposure (Martincorena, I. et al, 2015). Could this be observed in this study? 1) Martincorena, I. et al. High burden and pervasive positive selection of somatic mutations in normal human skin. Science 348, 880-886 (2015).

Response: We have checked for the dinucleotide mutations in our dataset, but did not find any. We discuss this now on p. 13.

Reviewer #3 (Remarks to the Author):

A-B. Summary of key results and originality or interest.

I believe the most novel and important result is the finding that UVR2 protects from UV-induced mutations especially in the germline after the split between male and female tissues. And that both gametes contribute a comparable amount of UV-induced mutations to the next generation. I think these are unexpected, well supported findings that will inspire future work and theory development. They are accompanied by many other results that may not make the same impact, because they pretty much confirm the expectations. For example, the relative importance of the pathways affected in UV-sensitive mutants is important, but does not pose further exciting questions by itself. In my opinion, the same is true about the spectrum and distribution of UV-induced mutations along the genome.

Response: We thank the reviewer #3 for the positive evaluation of our work.

C. Data & methodology

The dataset is excellent and I think the methods are appropriate. My only disappointment was the low resolution of the Luciferase-tagged UVR2 expression. To see the flower lightened in Figure 3b is compatible with the idea that most UV-induced mutations prevented by UVR2 would have happened in the germline after the split of male and female organs. I hoped that a closer look would have shown what parts of the flower express more UVR2. However, a higher resolution is not essential to hold the main result.

Response: We have now included more detailed analysis of the UVR2 reporter line in Figure 3b-e, Supplementary Figure 5 and the results section.

D. Use of statistics

I have only two minor concerns about statistics. One is the general use of Fisher's exact test. This test is used to check the independence between two categorical variables. Thus, I think it is appropriate to see if mutations appear more or less frequently than expected in methylated or non-methylated cytosines, for example. In several other cases, I agree it's well used. However, I cannot understand how Fisher's exact test is used to compare the normalized number of mutations per genome between two treatments or two genotypes.

Response: We discussed this issue with a specialist in statistics during the data analysis. His suggestion was to use the Fisher's exact test (or alternatively Chi-square test). For the comparison between two treatments or two genotypes, the categories were defined mutated and non-mutated. For example, for the *uvh1* genotype with the two categories control and UV-B (given dose) we would therefore have the following contingency matrix:

uvh1	Mutated positions*	Non-mutated positions*
Control	89	521,522,132
UV-B	120	521,547,031

*Mutated - all accepted mutations for the five *uvh1* genomes with the same treatment, and non-mutated - all accepted positions in the five sequenced genomes minus the mutated positions (values from Supplementary Tables - List of samples). A two-sided Fisher's exact test leads to a p-value of 0.03772 in this example, which is also reported as such in the manuscript.

My other request is to see, when possible, the p-value, rather than just saying it's significant at 0.05 level.

Response: We now included this information in the text and also provide more detailed classification in the figures: * $P < 0.05$, ** $P < 0.01$, *** $P < 0.001$.

E. Conclusions.

The reasoning behind the main conclusions is sound. From the 8:1 ratio of heterozygous to homozygous mutations in UV-treated uvr2 plants is natural to conclude that most mutations happen after the split between female and male cell lineages. The mating experiment and the conclusions from it are also correct.

Response: Thank you for recognizing the power of this genetic experiment.

F. Suggested improvements.

I encourage the authors to elaborate the implications of the main conclusions. How does it relate to what is known about the development of the flower? When do the male and female cell lineages split? The similar numbers of mutations in male and female cell lineages requires a better explanation than a "similar number of cell cycles leading to plant male and female gametes compared to animals". It is evident to the broad audience that there are many more pollen cells than ovules, suggesting a very different number of cell cycles.

Response: We have re-written this paragraph and included more detailed description on the number of DNA replications in plant microspore and megaspore development. While there is a good estimation for the number of DNA replications and cell divisions during and after meiosis, such information is missing for development of anthers and carpels.

If proper importance is given to these issues, some re-structuring of the text could improve readability and interest. It is unfortunate to find these nice results hidden behind lengthy descriptions of mutation counts and non-significant or mildly interesting comparisons. It may be possible to shorten it.

Response: We have now shortened specific parts based on the suggestions of all four reviewers.

Some details:

- Supplementary figure 4 is actually a table.

Response: We converted the original Supplementary Figures 4 and 6 to Supplementary Tables 3 and 4.

- On supplementary table 1, it seems that the mutations per site "per generation" are not given "per generation" when more than 1 generation, but the

accumulated after 2 or 3 generations. Just for clarity.

Response: Indeed. We corrected the description to “Cumulative number of mutations per haploid genome”. Thank you for noticing this.

G. References.

Honestly, I didn't check if there are some better references.

H. Clarity and context.

I don't think the abstract makes justice to the paper. It presents it as if we didn't know much about how plants protect themselves from UV radiation. But the genes involved are already described. The real contribution is the precise measurements of the numbers of mutations prevented by these pathways in different conditions, which provides theoreticians with evidence to test hypotheses about the evolution of plants and pathways.

Response: We agree that our formulation was not precise enough and modified it.

Reviewer #4 (Remarks to the Author):

This is an interesting paper with some potentially important conclusions. However, there are a number of serious concerns about the data and the validity of some of the analyses and conclusions.

Major Concerns:

1. The UV-B exposure experiments do not use an appropriate control. The UV treatments are additionally exposing the plants to high levels of UV-A (~75% of total UV) as well as to UV-B at different levels, but this UV-A is not present in the control treatments (Supp. Fig. 1A). The effects observed in the exposed lineages could thus be caused by this UV-A, rather than by the UV-B which the paper is supposed to be about. The experiments need a proper control (as for example is in the cited Ries et al., 2000 paper). Without this kind of control it is impossible to determine how much of what is seen is genuinely due to UV-B exposure.

Response: It is technically impossible to exclude UV-B only and include all UV-A radiation at the same time. This will also never happen in nature due to the ozone absorption. There is no filter glass existing at the moment, which transmission is zero below 315 nm (the defined border between UV-B and UV-A) and 100 % right above 315 nm. Even the small filter glasses (e.g. WG filters by Schott) cannot simulate such a situation exactly and, in addition, are too small in size for our facility to run such a big experiment. Therefore, in sun simulators we have to deal with different available big filter glasses and the plants are exposed to UV-B and UV-A are getting little more UV-A as plants exposed to UV-A only. We used different combinations of borosilicate, soda lime and float glass filters.

This is, to our knowledge, actual state of the art to adjust the short-wave cut-off of the UV and was also done this way by Ries et al. (Nature, 2000). We now made clear that whole UV-B radiation and short wavelength UV-A radiation were filtered out in the control treatment in the Results and the Materials and methods sections. It is not very likely that the mutation were caused by the short wavelength UV-A rather than UV-B as the later is more energetic and can also induce pyrimidine dimers more efficiently. There may be a complex and tissue-specific relationship between the numbers of UV-induced pyrimidine dimers (pre-mutagenic structures) and the final number of mutations. Therefore, we are not convinced that quantifying amounts of CPDs in e.g. leaf tissues will be directly comparable to the number of UV-B-induced mutations in the next generation (if this is what the reviewer is proposing).

2. The genome coverage ('accepted positions') is poor, in some cases less than 60% of the genome. This is well below the standard accepted for this kind of analysis (usually 90% or above), and means that many of the conclusions reached in the manuscript may not be robust. The samples need to be sequenced to better coverage.

Response: We can access on average 75% (min. 52.9%, max. 89%) of the 120 million positions in the genome under consideration. This is only slightly less than in a similar study (Ossowski et al., 2010; 77% to 80% accessible sites). However, Ossowski et al. reported just homozygous mutation, which are easier to call and therefore a coverage cutoff of 5x was assumed to be sufficient. A higher coverage threshold is needed to differentiate true heterozygous sites from variable sites due to technical artifacts (Wijnker et al. (elife), Supplementary Figures 6 and 7). However, we just look at ~5 to 10% less sites although using a 4 fold higher coverage cutoff. There is a recent similar study (Yang et al. (2015), Nature) in Arabidopsis following a similar approach to identify newly induced mutations. Unfortunately, they do not report the size of accessible space as we do, but they report an average raw read coverage of ~50x. This is lower than our average raw read coverage of 60x. They also report the fraction of genome covered by ≥ 5 uniquely mapped reads, which is approx. on average 95% (Yang et al, Supplementary Table 1). This is again in the range what we cover on average in this study (93%). However, since they apply similar criteria as us in their analysis, they probably also look for mutations in a smaller fraction of the genome. We would like to stress that our estimated mutation rate is not confounded by looking at a smaller fraction of the genome, since the mutation rate estimations as mutation frequency is estimated across the number of accessible sites (reflecting sites that could be scored as homozygous wild-type, heterozygous or homozygous mutant in the focal genome) and not across the entire genome (see Material and Methods for details).

3. The title is inaccurate. Natural UV-B is not studied in this work. '...trans-generational genome stability under simulated natural UV-B....' (or something similar) would have been better.

Response: We agree and included “simulated” into the title.

4. The Abstract claims that ‘...UVR2 photolyase is the major mechanism required for sustaining plant genome stability across generations’. This ‘major mechanism’ conclusion cannot be drawn from the work described in this manuscript. The only conclusion that can potentially be drawn (provided the observations are shown to be robust) is that UVR2 provides a mechanism for protecting against UV damage. Other mechanisms (e.g., mismatch repair) are likely as (if not more) important in ‘sustaining genome stability across generations’. The authors need to be more circumspect in the accuracy of their writing (a criticism that applies to throughout the entire manuscript, and not just in this specific instance).

Response: We modified this sentence as follows: “The results revealed that reversal of pyrimidine dimers by UVR2 photolyase is the major mechanism required for sustaining plant genome stability across generations under simulated natural UV-B.”

Additional concerns/comments:

1. Pg. 4 and Fig. 1b. The growth phenotypes shown are interesting. It makes sense that the tt4 mutation reduces photoprotection and hence growth in increased UV. But why should lack of UVR2 (a DNA repair enzyme) in uvr2 affect growth in increased UV? It is unlikely that this effect is due to somatic mutation as all such mutations would be heterozygous. Perhaps UVR2 has functions other than DNA repair? Could the authors comment?

Response: We agree that the reduced growth of *uvr2* under UV-B is not caused by the somatic mutations in the first irradiated generation because: (i) deleterious mutations are mostly recessive and thus should not be effective; (ii) the phenotype occurred consistently for all *uvr2* and *uvr2 uvr3* plants, which would require repeated mutagenesis of the same gene(s). Instead, the reduced plant growth may be caused by cell cycle delay and cell death. There is a good evidence that *UVR2* is transcriptionally regulated by specific E2F transcription factor DEL1, which also links to cell cycle and particularly endoreplication control (Radziejwoski et al., EMBO J, 2011). DEL1^{KO} plants compensate a stress-induced reduction in cell number by ploidy-dependent cell growth (faster continuation of endoreduplication).

2. Pg. 5. A false positive discovery estimate is provided. But we also need to know the false negative discovery rate.

Response: This is very good suggestion. Thank you. We prepared Supplementary Figure 6 and 7 and description in materials and methods showing how to tackle this problem. In short, mutation allele frequency distribution shows clearly separated group of homozygous mutations and partially overlapping peaks of heterozygous mutations and sequencing errors. We conducted a

simulation study in order to quantify the number of false positives and negatives in our study (please see Materials and methods for details).

3. Pg. 5. The authors report a spontaneous mutation rate which (in Discussion) they claim to be 'higher' than previous estimates. Is this difference statistically significant? If not, don't comment on it and don't say that it is higher. Actually this observation is based on a very small sample size (9, 13 and 10 mutations (Supp. Table 1) is not enough to draw meaningful spontaneous mutation rates), and suspect for that reason. But if the rate is genuinely higher, could it be due to the very high irradiance that the lineages are exposed to (340 $\mu\text{mol m}^{-2} \text{s}^{-1}$ is stressfully high; standard growth conditions are 100-150). Could this be a reason why, if indeed the spontaneous mutation rates are indeed robustly higher, they are 4-fold higher than those observed in the Ossowski et al. paper?

Response: Indeed, the PAR value of 340 $\mu\text{mol m}^{-2} \text{s}^{-1}$ used in our experiments is higher than the irradiance usually reached in standard growth chambers. However, the application of realistic UV-B doses efforts a PAR intensity of this range. The applied PAR is in fact much lower comparable to natural sunny conditions outside, where PAR of 1000 $\mu\text{mol m}^{-2} \text{s}^{-1}$ and more are measured frequently during spring and summer seasons. The PAR applied in our study is realistic e.g. for shady habitats, where plant perceive also UV. Other authors like Ries et al. (Nature, 2000) and Favory et al. (EMBO J, 2009) even used PAR up to 800 $\mu\text{mol m}^{-2} \text{s}^{-1}$ in their studies.

At this point, we cannot exclude that the higher mutation frequency in our control samples, when compared to other studies (e.g. Ossowski et al., Science, 2010), is due to higher PAR intensity. However, if this is true, then the spontaneous mutation frequency estimates in our study may be more realistic due to more natural growth conditions in the sun simulator.

4. Pg. 5. '...1.2-2.2-fold higher than the average number of mutations in control-treated plants, but not significantly higher than expected (Fisher's exact test, $\alpha = 0.05$) given the number of analysed genomes.' If it isn't significant it isn't higher and the authors shouldn't say that it is. This lack of statistical robustness is a thread running through the entire manuscript.

Response: We removed this statement and provide direct P value estimates now.

*5. Pg. 6. The authors suggest that UV-B-treatment of *uvr2* plants increases the frequency of non-synonymous mutations in protein coding regions. However it is not clear if this increase is simply the consequence of increased numbers of mutations throughout the genome or if the authors are claiming that mutations under these conditions are somehow targeted to protein coding regions. Clearer writing is needed here.*

Response: We have re-written and shortened this paragraph.

6. Pg. 7. The authors claim that transposons are prone to G:C to A:T transitions under UV-B-free conditions and that genes are not. This is a very surprising finding. Why was it not found during the analyses described in the Ossowski et al. paper? Have the authors re-analysed the data in the Ossowski et al. paper (they should, because there are more mutations there than in the data in the paper currently under review) to see if they find the same thing? Perhaps this is another case of over-interpreting from insufficient data (insufficient mutations)?

Response: Thank you for this comment. We re-analyzed the Ossowski et al. (2010) data in the same way as our data, and included the result as Supplementary Figure 4c. Ossowski et al. reported a total of 99 single base pair substitutions (58 C to T) and we found a total of 129 single base pair substitutions (68 C to T) under control (UV-B free) conditions. Therefore the power of both datasets on this subject should be comparable. Analyzing the Ossowski data revealed a very similar picture as compared to our data. Ossowski data showed more GC to AT transitions in TEs (65% of all substitutions) compared to genes (42% of all substitutions). In addition, also here we found more mutations in the TE space as compared to the gene space. This effect was even stronger if just C to T changes were considered, as expected if 5-methyl cytosine de-amination was the main cause for a mutation. The minor differences could be due to version differences in the annotation (TAIR8 vs. TAIR10), and much shorter read length, which change the accessible space in the reference genome.

7. Fig. 1b, c. Why draw the histograms in this complicated way (the boxes with individual data points rather strangely arranged). Conventional means plus/minus SE would show all that you need to show.

Response: We discussed the graph format extensively among the co-authors prior to the first submission. From several variants (including the one proposed by the reviewer #4), we selected the current format as the most informative one. Therefore, we would like to keep it.

8. Fig. 2a. What is frequency (what are the units?)? Also it is not clear what we are looking at here. It needs to be absolutely clear that this is not just data from WT plants but is actually data from WT plus mutant lines carrying UVR2 (but not homozygous for uvr2).

Response: We changed the description to percentage in Fig. 2a and added information on the composition of the compared groups to every figure. In addition, we included detailed overview of genomes used for individual experiments in Supplementary Table 1.

9. Fig. 3a. 'Homozygous/heterozygous mutation...' in the column should be corrected as 'Heterozygous/homozygous mutation...'

Response: Corrected.

Reviewers' comments:

Reviewer #1 (Remarks to the Author):

The is a review of a revised ms- please see my earlier comments on content and significance.

This paper now looks good to me, except for some writing at the top of page 12:

P12 "(6- 4)PPs are repaired by both transcription-coupled repair and global genomic repair NER pathways, while CPDs are resolved only by global genomic repair in humans³⁴⁻³⁵. Currently it is unknown whether this may be due to absence of DNA photolyases in humans³⁶, but it is plausible to speculate that the weak *uvr3* and *uvh1* Arabidopsis phenotypes may be caused by a partial redundancy of both pathways in (6-4)PPs repair."

First, humans do not repair global CPDs effectively, in contrast to 6-4s. The observed repair of CPDs in humans is mostly transcription-coupled. Or perhaps the authors are trying to say that only humans perform global CPD repair? But they don't. Though many other organisms do. Second, I'm not sure why we're discussing humans at all. Human repair capacity doesn't explain mutation rate in Arabidopsis. Both global and transcription coupled repair of CPDs has been characterized in Arabidopsis. Global repair of 6-4s, and its efficiency vs global repair of CPDs, has also been described. I would suggest discussing this instead.

Reviewer #2 (Remarks to the Author):

This manuscript focus on Arabidopsis mutation rates variation affected by UV-B radiation and UV response genes. The major effect of UVR2 on preventing UV mutations is of great interest. Most of our previous questions have been answered. But I still have some concerns after the revision.

1. Mutation validation needs not only to prove the presence in the progenies but also the absence in the parents. But the validation in parents are still missing in this paper. Also the inheritance of these mutations haven't been validated to confirm they are homogeneous or not.

2. Of the 900 in-silico simulated mutations, 685 located in accessible sites and 684 were identified. Only 76% mutations (685/900) are accessible. Does this mean that some mutations have been missed as the criteria is too strict?

3. The mutation rate in this paper is much higher than that (7×10^{-9}) in Ossowski et al (2010) and others. Is it possibly an overestimation by sample size or other conditions in this paper or it's an underestimation in the previous study? This may not change the fact that UV radiation and UV response genes affect mutation rates but may confuse others in the mutation rates of Arabidopsis.

This work is of great value, but I think some revisions are still necessary.

Reviewer #3 (Remarks to the Author):

The authors replied to all my requests, and explained what I did not understand before. I see the improvements reflected in their manuscript, and I recommend its publication.

However, an overall impression of low readability did not vanish completely. The results include a series of statistical comparisons that do not flow easily. A number of variables modulate the effect of UV-B on mutation rate: genetic background, methylation, sequence context... Maybe the inclusion of all the variables in a single model would offer a broad picture and help separate the effects of each factor. I wonder if it is possible at all to fit a generalized linear model, for example.

In my effort to integrate the information coming from all these comparisons, I understood the following.

The higher mutability of methylated cytosines explains both the high fraction of G:C->A:T transitions among all observed mutations, and the overrepresentation of mutations among transposable elements, which are more methylated, both in control and UV-B conditions. But it is still unclear why UV-B treatment causes a higher increase in G:C->A:T transitions in genes than in other genomic regions. In addition, non-synonymous positions seem to be more sensitive to UV-B treatment than synonymous positions. These patterns are not explained by methylation levels. Could they be explained by a higher frequency of the UV-B-sensitive TC(C/T) sequence motif in genes and in non-synonymous sites? I don't see this addressed anywhere, and I don't think it would be very difficult to check.

Finally, two details. First, on page 7, line 153, "all other types of substitutions were significantly reduced". I understand the authors mean the frequencies or proportions of all other types of substitutions were reduced, relative to the total number of mutations. Unless stated explicitly, the sentence seems to suggest that UV-B treatment reduces the number of mutations. And second, on page 14, line 336, note the typo in "could be transmitted to the in future generations".

--Summary of additional comments from reviewer #2 and reviewer #3 (regarding response to reviewer #4):

Reviewer #2 felt that major concern 2 of reviewer #4 required a more thorough response. Reviewer #2 felt that it remained unclear whether the average percentage of accessible sites was enough and the potential effect of missing mutations on the conclusions.

Reviewer #3 commented on the potential contribution of UV-A to the results. Reviewer #3 indicated that it was not technically possible to exclude UV-A from the treatment but felt that this inherent limitation be more clearly acknowledged. Reviewer #3 agreed that while it was likely that the majority of the mutations were UV-B induced, it was somewhat contradictory to neglect the possible influence of UV-A, even if indirect, while suggesting on

page 11 (line 258) that UV-A may be indirectly responsible for higher mutation rate in the control. Reviewer #3 suggested more careful wording so that instead of referring to "UV-B treated plants" the phrases "UV-A and UV-B treated plants" and "UV-B or UV-A induced mutations" should be used. Reviewer #3 felt that Ries et al 2000 was not entirely comparable as Ries et al used varying amounts of UV-B.

Regarding comment number 8 by reviewer #4, reviewer #3 felt it was important to make clear that the proportions of the different types of mutations are not referred to as mutation rates. In this regard, reviewer #3 felt the legend to figure 2 should not refer to 'frequency' which may be misunderstood.

Reviewer #3 also pointed out a typo "Ossowski et al. 201022" in figure 2 and commented that the vertical axis of figure 1a were not correctly ordered.

Response to the reviewer's comments

UVR2 ensures trans-generational genome stability under simulated natural UV-B in *Arabidopsis thaliana* (Manuscript: NCOMMS-16-00793A)

Reviewers' comments:

Reviewer #1 (Remarks to the Author):

The is a review of a revised ms- please see my earlier comments on content and significance.

This paper now looks good to me, except for some writing at the top of page 12:

P12 "(6- 4)PPs are repaired by both transcription-coupled repair and global genomic repair NER pathways, while CPDs are resolved only by global genomic repair in humans³⁴⁻³⁵. Currently it is unknown whether this may be due to absence of DNA photolyases in humans³⁶, but it is plausible to speculate that the weak *uvr3* and *uvh1* *Arabidopsis* phenotypes may be caused by a partial redundancy of both pathways in (6-4)PPs repair."

First, humans do not repair global CPDs effectively, in contrast to 6-4s. The observed repair of CPDs in mostly transcription-coupled. Or perhaps the authors are trying to say that only humans perform global CPD repair? But they don't. Though many other organisms do.

Second, I'm not sure why we're discussing humans at all. Human repair capacity doesn't explain mutation rate in *Arabidopsis*. Both global and transcription coupled repair of CPDs has been characterized in *Arabidopsis*. Global repair of 6-4s, and its efficiency vs global repair of CPDs, has also been described. I would suggest discussing this instead.

Response: Thank you for pointing this out. We agree that the comparison with human data was not the most direct. Now we compare our data to the *Arabidopsis* data derived from the Britt lab in the early 90' on page 12.

Reviewer #2 (Remarks to the Author):

This manuscript focus on *Arabidopsis* mutation rates variation affected by UV-B radiation and UV response genes. The major effect of UVR2 on preventing UV mutations is of great interest. Most of our previous questions have been answered. But I still have some concerns after the revision.

1. Mutation validation needs not only to prove the presence in the progenies but also the absence in the parents. But the validation in parents are still missing in this paper. Also the inheritance of these mutations haven't been validated to confirm they are homogeneous or not.

Response: We analyzed at least 10 genomes per genotype, which were offspring of the same ("grandparent") plant in order to identify common polymorphisms in each genomic background. This reconstructs in large parts the genome of the original plant. Polymorphic sites that are shared in these genomes represent differences already present in the original genome. Homozygous mutations are present in each of these lines and are therefore easily identified as mutations that existed before the beginning of the experiment. Heterozygous differences in the parental genome can be lost in the

next generation and therefore appear in only a few of the offspring plants. The probability of a heterozygous mutation to be lost over the subsequent generations is 0.5. Therefore, there is a negligible probability of 0.5^9 that a heterozygous polymorphic site is only present in one of the 10 genomes and thus being scored as new mutation in our analysis although it was already present in the original plant. To check the degree of inheritance of the reported mutations, we analyzed the generation G2 and G3 of the *uvr2 uvr3* mutants under UV-B conditions. Plants sequenced as G2 and G3 are actually not mother and daughter plants, but are “aunt and nieces” (Figure S1c). Therefore, the degree of kinship between these generations should be 25%. In the G2 plants 1, 2, 3, 4 and 5, we scored 114, 86, 94, 104 and 103 newly induced mutation, respectively, and re-identified 43, 20, 25, 22 and 21 of these mutation also in the plants in G3, implying an average kinship of 25.8%, which is very close to the expected value of 25%. Therefore, we are confident that the vast majority of mutations reported with our pipeline are real and inherited to the next generations.

2. Of the 900 in-silico simulated mutations, 685 located in accessible sites and 684 were identified. Only 76% mutations (685/900) are accessible. Does this mean that some mutations have been missed as the criteria is too strict?

Response: Please note, it is a common fact for all resequencing efforts that not 100% of the genome is analyzed (and thus there will never be the chance to find all mutations). The percent of accessible sites is defined by stringency that is applied to the base calling. More relaxed base calling allows analyzing more of the genomic regions (as some regions are not so well supported by the alignments as other regions); however, this comes at the costs of an increased error rate. In this example, the mutation rate is estimated based on the 684 mutations considering the space in which only 685 mutations could have been identified. Therefore the mutation rate is highly accurate even if not calculated “using the whole genome”.

As long as sufficient space of the genome is analyzed and as long as this space represents the entire genome (what we have shown is the case) limiting the analysis to parts of the genome does not impact on the estimated mutation rates. We have analyzed around 3/4 of each mutant genome, which is around 90 million sites per genome, which implies more than sufficient power. The reason to be restrictive was to exclude as many false positive mutations as possible.

To make this point more clear, we conducted additional simulations now inducing artificial mutations (which can be homozygous or heterozygous by equal chances) every 5000bp into the reference sequence (23,795 in total), implying a rather high mutation rate of 0.0001997 (for all detailed numbers please see Table Sa). We simulated an average coverage of 40x, which is in the lower range of average coverage in our real data. Applying the same “stringent” criteria as in our study, we find again that we can identify 77% (18,445 of 23,795) of the mutations in 77% (91,660,765 of 119,146,348 are accessible) of the genome, reporting no false positives and leading to a mutation rate estimate of 0.0002012, which is very close to the actual mutation rate. Since we normalized our mutation rates with the number of accessible sites, analyzing only parts of the genome did not skew our estimate. For more information please see also results of the simulation below.

Table Sa.

	# mutations	# FP (cutoff 0.3)	# FN (cutoff 0.3)	# accessible sites	mutation rate	log fold change to true mutation rate	% mutations	% accessible sites
simulated data	23,795	0	0	119,146,348	0.0001997		100%	100%
45x simulated, cutoff 20x	18,445	0	81	91,660,765	0.0002012	0.010929835	77.5	76.9
20x simulated cutoff 10x	21,456	55	478	102,314,390	0.0002097	0.070448352	90.2	85.9
20x simulated, cutoff 15x	20,786	7	438	99,896,375	0.0002081	0.059184202	87.4	83.8
20x simulated, cutoff 20x	16,287	0	272	81,264,504	0.0002004	0.005099935	68.4	68.2

Lowering the coverage cutoff to 10x, in order to have access to larger proportion of the genome, would lead to a higher number of false positives in our set of reported mutations severely affecting the mutation rate estimate especially for the genomes with very low number of mutations (<10). The number of false positives is independent from the real number of new mutations in the genome, but only depends on technical problems like sequencing and alignment errors, whereas the number of false negatives strongly depends on the number of mutations in the genome.

To show this we simulated an additional dataset on the same genome. So far, none of the short read simulators is able to mirror the unequal coverage of sites in the genome found analyzing real data. Figure Sn shows the distribution of coverage in the complete genome in real data with an average sequencing depth of 45x (black). So far, we simulated data with an average coverage of 40x, which is even lower, but you see a much more even coverage along the genome, which is almost for all sites > 20x (blue). In order to show that we increase the number of false positive by lowering the coverage cutoff, we simulated a dataset with an average sequencing depth of 20x in order to have more sites in the simulated genome below 20x (red). We now analyzed the data with the same pipeline just applying different coverage cutoffs (see Table Sa for all the numbers). With 10x coverage cutoff, we found 21,456 mutations including 55 (0.2%) false positives in 102,314,390 (86%) accessible sites. This would lead to a mutation rate estimate of 0.000210, which is a slight overestimate of the simulated mutation rate (lfc = 0.07). However, if the real mutation rate was much lower (e.g. 8-e8 corresponding to 10 mutations in the Arabidopsis genome), we still would find 55 false positives and severely over estimate the mutation rate. Increasing the cutoff to 15x reduces the number of false positives already to 7 (0.03%) reporting 20,786 mutations in 99,896,375 accessible sites (84%) leading to a better mutation rate estimate of 0.000208 (lfc = 0.059). However, with 10 mutations also this number of false positives would lead to a severe overestimate. Going to 20x coverage cutoff leads to no false positives, while identifying 16,287 (68%) mutations in 81,264,504 accessible sites (68%) leading to a mutation rate of 0.00200 (lfc = 0.005), which is the one with the lowest error. Considering the case with just 10 mutations under sampling the genome (accessible sites) is a critical, because we could miss by chance all 10 mutations and estimate a mutation rate of 0. However, in our study we can access 75% of each genome on average and we sequenced 5 plants for each combination leading to a very low probability ($0.25^5 < 0.0001$) in the case of just 1 mutation per genome that we miss the one mutation in every genome. At the same time, our cutoffs ensure that we report not severely upwardly biased mutations rates due to false positives.

Figure Sn.

3. The mutation rate in this paper is much higher than that (7×10^{-9}) in Ossowski et al(2010) and others. Is it possibly an overestimation by sample size or other conditions in this paper or it's an

underestimation in the previous study? This may not change the fact that UV radiation and UV response genes affect mutation rates but may confuse others in the mutation rates of Arabidopsis.

Response: We find on average 1.4 newly introduced mutations per haploid genome in the Col-0 background, which is very unlikely an overestimation due to the very low false positive rate as described above. The mutation rate in Ossowski et. al (2010); however, might have been a conservative estimate. For example later re-sequencing of one of the mutation accumulation lines (Becker et al. (2011)) revealed a mutation (affecting the genome-wide levels of DNA methylation) that has not been reported in the original paper. Due to short reads of lower quality some mutations might have been missed potentially leading to a slightly lower estimate of the mutation rate.

This work is of great value, but I think some revisions are still necessary.

Reviewer #3 (Remarks to the Author):

The authors replied to all my requests, and explained what I did not understand before. I see the improvements reflected in their manuscript, and I recommend its publication.

However, an overall impression of low readability did not vanish completely. The results include a series of statistical comparisons that do not flow easily. A number of variables modulate the effect of UV-B on mutation rate: genetic background, methylation, sequence context... Maybe the inclusion of all the variables in a single model would offer a broad picture and help separate the effects of each factor. I wonder if it is possible at all to fit a generalized linear model, for example.

In my effort to integrate the information coming from all these comparisons, I understood the following.

Response: Building a generalized linear model just considering UV treatment and genomic background as independent variables and mutation rates as response variable would be easily possible. However, we think that it is quite clear that the lack of UVR2 with UV+ treatment has the largest impact on the mutation rate and a generalized linear model would lead to the same result. Adding methylation and sequence context into the model has two major problems. The first one is the very small number of observations for some of the genomic background and treatment combinations. The methylation data we used does not reveal for every mutated C whether it is methylated or not, in addition to anyway very low mutation rates in some genomes, which is the reason why we pooled the data according to UV- and UV+. Secondly, both variables, DNA methylation and sequence context, would not be independent to the response variable (mutation rate). They would be fractions of the number of mutated sites and to make them independent would require methylation data from each plant sequenced to determine independently the number of methylated cytosines mutated and not mutated. However, this is out of scope of this study but might be an interesting aspect for future studies on this topic. The sequence context will always dependent since the background genome and therefore the background frequencies remain always the same. However, as an alternative, we prepared a graphical representation (Figure 4) in order to summarize the major conclusions of the manuscript.

The higher mutability of methylated cytosines explains both the high fraction of G:C->A:T transitions among all observed mutations, and the overrepresentation of mutations among transposable elements, which are more methylated, both in control and UV-B conditions. But it is still unclear why UV-B treatment causes a higher increase in G:C->A:T transitions in genes than in other genomic regions. In addition, non-synonymous positions seem to be more sensitive to UV-B treatment than synonymous positions. These patterns are not explained by methylation levels. Could they be explained by a higher frequency of the UV-B-sensitive TC(C/T) sequence motif in genes and in non-

synonymous sites? I don't see this addressed anywhere, and I don't think it would be very difficult to check.

Response: We calculated genome-wide UV-B sensitive triplet frequency in different genomic contexts (gene, TE, intergenic). Frequencies of the different triplets in genes was on average 5% higher than the background frequency (~55% compared to ~50%) (Figure Sc, Figure Sm). However, it does not explain the 20% increase of UV-B induced mutation in genes compared to the control.

Figure Sc.

Figure Sm.

In order to check if UV-B induced mutation in genes are enriched for non-synonymous changes, we extracted the codons with a UV-B sensitive context at the first, second or third position of all protein coding genes in *A. thaliana* (UV-B sensitive = a C on the plus or minus strand with a pyrimidine before and/or after). Of all codons 40% and 32% carry a C in a UV-B sensitive context at the first and/or second position, respectively. Almost all (95% and 99%) potential C->T transitions would lead to a non-synonymous change. At the third codon position 32% show a UV-B sensitive context, but here only the minority of C->T changes would lead to a non-synonymous change (9%).

Hence, of all UV-B sensitive codon position, a C->T transition would in 70% of the cases lead to a non-synonymous change. These numbers remain about the same if we just consider the enriched TC(C/T) context. From these numbers we would expect a ratio of almost 2.3:1 for non-synonymous to synonymous changes and indeed we find of 395 potentially UV-B induced mutations in coding regions 292 non-synonymous and 103 synonymous changes reflecting a 2.8:1 ratio. This difference is not significant (fisher exact test $p > 0.07$) implying that the number of observed non-synonymous changes is not significantly larger than expected.

Finally, two details. First, on page 7, line 153, "all other types of substitutions were significantly reduced". I understand the authors mean the frequencies or proportions of all other types of substitutions were reduced, relative to the total number of mutations. Unless stated explicitly, the sentence seems to suggest that UV-B treatment reduces the number of mutations. And second, on page 14, line 336, note the typo in "could be transmitted to the in future generations".

Response: Corrected.

--Summary of additional comments from reviewer #2 and reviewer #3 (regarding response to reviewer #4):

Reviewer #2 felt that major concern 2 of reviewer #4 required a more thorough response. Reviewer #2 felt that it remained unclear whether the average percentage of accessible sites was enough and the potential effect of missing mutations on the conclusions.

Response: We have addressed this in detail in response to the reviewer #2.

Reviewer #3 commented on the potential contribution of UV-A to the results. Reviewer #3 indicated that it was not technically possible to exclude UV-A from the treatment but felt that this inherent limitation be more clearly acknowledged. Reviewer #3 agreed that while it was likely that the majority of the mutations were UV-B induced, it was somewhat contradictory to neglect the possible influence of UV-A, even if indirect, while suggesting on page 11 (line 258) that UV-A may be indirectly responsible for higher mutation rate in the control. Reviewer #3 suggested more careful wording so that instead of referring to "UV-B treated plants" the phrases "UV-A and UV-B treated plants" and "UV-B or UV-A induced mutations" should be used. Reviewer #3 felt that Ries et al 2000 was not entirely comparable as Ries et al used varying amounts of UV-B.

Response: Control treatment does not contain UV-B, but a certain amount of UV-A compared to the "UV-B-treatment" •depending on wavelength (due to filter characteristics): 10% at 330 nm and up to 80% and more from 360 nm onwards. Therefore, the difference between both treatments regarding the UV-A is only apparent in shorter UV-A wavelengths. So, we do not favor (after thorough discussion among the co-authors) changing the name to "UV-B and UV-A induced mutations". We now made the technical aspects of the treatment clearer in the manuscript on page 4 and in the materials and methods on page 15.

Yes, the amount of UV-A present in the control treatment may be responsible for the higher mutation rate compared to the other studies (Ossowski et al. and Yang et al.), but there is no comparison to our UV-B treatment at this point. We feel that this is clear from the context of the text.

There is most likely confusion concerning the Ries papers. There are two publications from the same author published in 2000. In the first one (Ries et al, PNAS, 97:13425-13429, 2000) the applied UV-B conditions differ from those applied in our study, while the treatments performed in the second publication (Ries et al., Nature, 406:98-101) are more similar to our work and the experiments were performed in the same sun simulator facility. But even the first paper was performed using filters very similar to our glass, showing similar transmission as shown in Ries et al. PNAS 2000, figure 1A.

Regarding comment number 8 by reviewer #4, reviewer #3 felt it was important to make clear that the proportions of the different types of mutations are not referred to as mutation rates. In this regard, reviewer #3 felt the legend to figure 2 should not refer to 'frequency' which may be misunderstood.

Response: Corrected.

Reviewer #3 also pointed out a typo "Ossowski et al. 201022" in figure 2 and commented that the vertical axis of figure 1a were not correctly ordered.

Response: Corrected.

REVIEWERS' COMMENTS:

Reviewer #2 (Remarks to the Author):

The authors have answered all my questions, and in principle they are acceptable. I recommend its publication.

The authors have checked the inheritance of mutations, which make them more convictive. The explanation of mutation standard is quite well. I still doubt that an overestimation of Arabidopsis mutation rate may exist, but this does not weak the authors basic conclusion.

Reviewer #3 (Remarks to the Author):

I think the manuscript is ready to be published. I don't have additional comments.